# Deep-learning-based three-dimensional label-free tracking and analysis of immunological synapses of CAR-T cells

**Moosung Lee[1,2†], Young-Ho Lee[3,4†], Jinyeop Song[1,2†], Geon Kim[1,2], YoungJu Jo[1,2], HyunSeok Min[5], Chan Hyuk Kim[3]\*, YongKeun Park[1,2]\***

[1]Department of Physics, Korea Advanced Institute of Science and Technology (KAIST), Daejeon, Republic of Korea; [2]KAIST Institute for Health Science and Technology, Daejeon, Republic of Korea; [3]Department of Biological Sciences, Korea Advanced Institute of Science and Technology, Daejeon, Republic of Korea; [4]Curocell Inc, Daejeon, Republic of Korea; [5]Tomocube Inc, Daejeon, Republic of Korea

**Abstract** The immunological synapse (IS) is a cell-cell junction between a T cell and a professional antigen-presenting cell. Since the IS formation is a critical step for the initiation of an antigen-specific immune response, various live-cell imaging techniques, most of which rely on fluorescence microscopy, have been used to study the dynamics of IS. However, the inherent limitations associated with the fluorescence-based imaging, such as photo-bleaching and photo-toxicity, prevent the long-term assessment of dynamic changes of IS with high frequency. Here, we propose and experimentally validate a label-free, volumetric, and automated assessment method for IS dynamics using a combinational approach of optical diffraction tomography and deep learning-based segmentation. The proposed method enables an automatic and quantitative spatiotemporal analysis of IS kinetics of morphological and biochemical parameters associated with IS dynamics, providing a new option for immunological research.

**\*For correspondence:**
kimchanhyuk@kaist.ac.kr (CHK);
yk.park@kaist.ac.kr (YKP)

[†]These authors contributed equally to this work

## Introduction

Understanding the immune response at the cellular scale requires knowledge regarding interactions between immune cells and their microenvironment. For T lymphocyte, which is one of the major immune cell types involved in adaptive immune responses, it communicates with antigen-presenting target cells via the formation of a nanoscale cell-cell junction, which is called the immunological synapse (IS). Specifically, the engagement of T cell receptor (TCR) by peptide-loaded MHC complex (pMHC) presented on the target cells leads to the formation of IS that coordinates the downstream signaling events required for the initial activation of T cells (*Bromley et al., 2001*; *Basu and Huse, 2017*). Previous studies have shown that the TCR-mediated IS comprises segregated concentric rings of supramolecular activating clusters, which are crucial for stabilization of the IS as well as the secretion of lytic granules (*Basu and Huse, 2017*; *Monks et al., 1998*).

Alternatively, recent studies have highlighted the IS structures formed by chimeric antigen receptor (CAR), a synthetic fusion protein comprised of an extracellular targeting and hinge domain, a transmembrane domain, and intracellular signaling domains (*Lee and Kim, 2019*; *Mukherjee et al., 2017*). The extracellular targeting domain of CAR is typically adopted single-chain variable fragment (scFv) of monoclonal antibodies, allowing the CAR-T cells to recognize various types of surface antigens independent of its MHC restriction. In particular, CD19-specific CAR-T cells have demonstrated remarkable anti-cancer efficacy in patients with B-cell malignancies (*van der Stegen et al., 2015*). Although CAR/antigen and TCR/pMHC complexes have different IS structures (*Davenport et al.,*

*2018*), the distinct dynamics and mechanochemical properties of the IS driven by these complexes remain understudied.

A variety of imaging techniques have been used to reveal the hierarchical details of the IS structures and their relevant functions. For instance, electron microscopy and single-molecule localization microscopy have resolved the spatial distributions of subcellular IS compartments beyond optical diffraction limits (*Choudhuri et al., 2014*; *Hu et al., 2016*). However, assessing dynamical changes in IS formation requires rapid and continuous imaging of immune cells. Fluorescence microscopy is useful in this regard (*Balagopalan et al., 2011*). Such fluorescence-based techniques have the advantage of chemical specificity. However, they are limited due to photo-bleaching and photo-toxicity, which necessitates the use of complementary label-free, rapid three-dimensional (3D) microscopy methods to assess long-term dynamic changes in IS morphologies (*Skylaki et al., 2016*). Recently developed lattice light-sheet microscopy enables long-term high-speed volumetric imaging (*Chen et al., 2014*), and thus can be potentially used for the study of the dynamics of CAR-T and the structures of IS. However, fluorescence labeling process requires complicated and time-consuming sample preparation, especially when multiplex fluorescence imaging is needed.

The development of label-free IS imaging has been limited to phase-contrast and differential interference contrast microscopy. The aim is to develop quantitative phase imaging (QPI) as a quantitative label-free imaging method to studying IS (*Park et al., 2018a*). Optical diffraction tomography (ODT) is a promising 3D QPI technique for imaging the 3D refractive index (RI) distribution of cells at a sub-micrometer spatial resolution (*Kim et al., 2016a*). Unlike nonlinear scanning microscopy that requires a long acquisition time due to weak signal intensities (*Zumbusch et al., 1999*; *Freudiger et al., 2008*; *Squier et al., 1998*; *Zipfel et al., 2003*), ODT enables fast 3D imaging via holographic recording. Also, because the reconstructed RI profile correlates with total cellular protein densities, ODT enables quantitative, photobleaching-free analyses of cell dynamics.

ODT has been actively used to study single-cell morphology (*Yoon et al., 2017*; *Kim et al., 2018*; *Park et al., 2008*; *Yang et al., 2017*). However, it has not yet been used to study cell–cell interactions that include immune responses. One of the primary reasons is the lack of an accurate 3D segmentation framework to distinguish interacting cell-to-cell interfaces, which is also a problem with other microscopy methods (*Uchida, 2013*). Manual marking is the most primitive segmentation strategy. It is effective but is too laborious and difficult for time-resolved volumetric segmentation. To overcome this barrier, automatic segmentation has been developed based on basic algorithms that include intensity thresholding, filtering, morphological operations, region accumulation, and deformable models (*Dimopoulos et al., 2014*). However, these methods often result in poor segmentation, particularly for adjoining cell segmentations, which occur in immune responses. To accurately and precisely segment immunologically interacting cells in an automated manner, a novel computational framework is needed.

Here, we present DeepIS, a computational framework for the systematic, label-free analysis of 3D IS dynamics of immune cells in ODT. Our framework is based on deep convolutional neural network (DCNN) that distinguishes adjoining immune cells, target cells, and IS surfaces from the obtained RI tomogram. The proposed framework enables the general, high-throughput, and automated segmentation of more than 1000 immune-target cell pairs. To validate the method, we applied this method to study the dynamics of CAR/antigen-mediated IS formed between CD19-specific CAR-engineered T cells (CART19) and CD19-positive K562 cancer cells (K562-CD19). The combined use of high-speed imaging capability of ODT enabled 3D high-speed CAR IS tracking in which a tomogram was measured every 3 to 8 s for a prolonged period of time (300 s to 10 min depending on the cell type). Exploiting the linear proportion between RI and protein density (*Barer et al., 1953*), we also demonstrate quantitative analyses of CAR IS kinetics using the morphological and biochemical properties. The results suggest that DeepIS offers a new analytical approach to immunological research.

## Results

### 3D time-lapse RI measurement of the CART19 and K562-CD19 cell conjugates using optical diffraction tomography

In order to perform ODT experiments in our study, we employed an experimental setup which is based on off-axis holography equipped with a high-speed illumination scanner using a digital micro-mirror device (DMD; DLP6500FLQ DLP 0.65 1080 p Type A DMD, Texas Instrument) (*Figure 1*). The setup enables the high-speed acquisition of a single tomogram within 500 milliseconds (*Figure 1a*; *Shin et al., 2015*; *Lee et al., 2017*). A 1 × 2 single-mode FC/APC fiber coupler was utilized to split a coherent, monochromatic laser (λ = 532 nm) into a sample and reference arms. The DMD was then placed onto the sample plane of the sample arm to control the illumination angle of the first-order diffracted beam striking the sample. To scan the illuminations at large tilt angles, a 4-*f* array consisting of a tube lens (Lens 1, *f* = 250 mm) and a condenser objective (UPLASAPO 60XW, Olympus Inc, Japan) magnified the illumination angle. The light scattered by live cells in a live-cell chamber (Tomo-Chamber, Tomocube Inc, Republic of Korea) was then transmitted through the other 4 *f* array formed by an objective lens (UPLASAPO 60XW, Olympus Inc, Japan) and a tube lens (Lens 2, *f* = 175 mm). The sample beam was combined with the reference beam by a beam splitter and filtered by a linear polarizer. The resultant off-axis hologram was then recorded by a CMOS camera (FL3-U3-13Y3M-C, FLIR Systems, Inc, USA) synchronized with the DMD to record 49 holograms of the sample illuminated at different angles. Using a phase-retrieval algorithm, the amplitude and phase images of the 1:1 conjugate between a CART19 and K562-CD19 cell were retrieved from the measured holograms (*Figure 1b*). Based on the Fourier diffraction theorem with Rytov approximation (*Wolf, 1969*; *Devaney, 1981*), the 3D RI tomogram of the sample was reconstructed from the retrieved amplitude and phase images (*Figure 1c*). To fill the uncollected side scattering signals due to the limited numerical apertures of objective lenses, a regularization algorithm based on the non-negative constraint was used (*Lim et al., 2015*). The maximum theoretical resolutions of the ODT system were respectively 125 nm laterally and 471 nm axially, according to the Lauer criterion defined as the Nyquist sampling period (*Lauer, 2002*; *Park et al., 2018b*). Note that the empirical spatial resolution of ODT has been known to be between the Nyquist sampling period and the Abbe criterion (*Simon et al., 2017*), so the Nyquist sampling period was used to set the lowest bound of the resolution. Finally, we approximated the protein densities from the reconstructed RI values using a RI increment per protein concentration, $\alpha = dRI/dc = 0.185$ mL/g[25]. We assumed this RI increment to be constant over cells for two reasons: (1) as in *Barer et al., 1953*, most protein molecules have narrow ranges of RI incremental values from 0.179 to 0.195 mL/g, (2) lymphocytes contain lipid-rich environment localized mostly on a 4-nm-thick membrane site, whose size is beyond

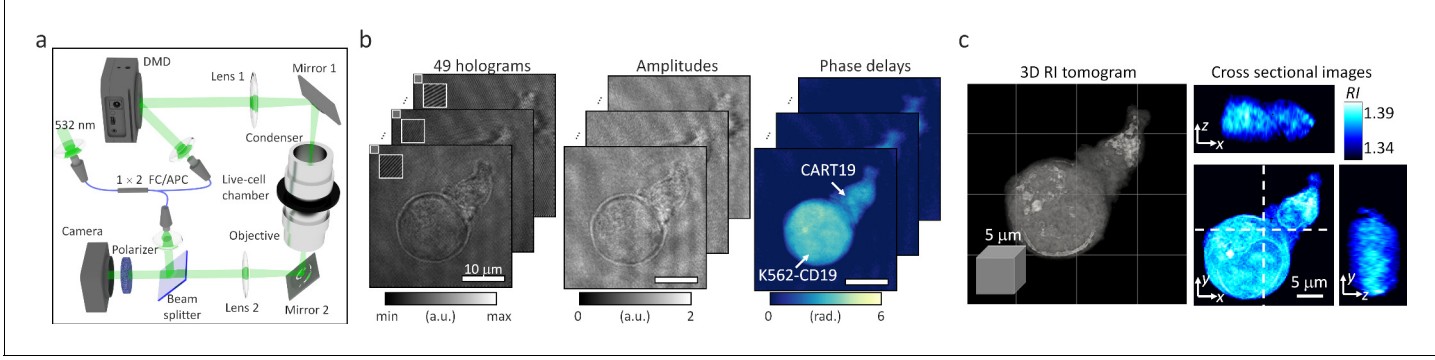

**Figure 1.** Data acquisition in optical diffraction tomography (ODT). (**a**) The experimental setup for ODT is based on a digital micro-mirror device (DMD) for high-speed illumination scanning. (**b**) Forty-nine holograms of 1:1 conjugate between a CART19 and a K562-CD19 cell were recorded at various illumination angles, and their amplitude and phase delay distributions were retrieved. (**c**) A reconstructed refractive index (RI) map.

The online version of this article includes the following video for figure 1:

**Figure 1—video 1.** Eight-hour time-lapse ODT imaging did not indicate photobleaching or signal losses.

https://elifesciences.org/articles/49023#fig1video1

optical resolution and thus the relatively voluminous adjacent cytoplasm would contribute the reconstructed RI tomogram rather than sub-resolved membrane layers. The average illumination intensity was 11.46 mW/cm$^2$, at which the samples did not suffer from phototoxicity or signal losses during long-term assessment (*Figure 1—video 1*).

## DeepIS establishment for automated assessment of CART19 IS dynamics

The IS tracking analysis is preceded by segmentation, which involves dividing volumetric sections for background, cell domains, and IS in the ODT RI map. However, iteration is required for parameter tuning of the manual segmentation method to obtain a single well-segmented label, which is prohibitive to obtain a dynamic dataset. Therefore, we established DeepIS framework based on the DCNN supervised learning method to enable general, high-throughput, and automated segmentation for 3D RI tomograms (*Figure 2*). The framework was developed in the following order: (i) Dataset preparation, (ii) Training stage, and (iii) Inference stage. These are detailed next.

### Dataset preparation

The preliminary step of supervised learning of DCNN is to prepare an annotated dataset (*Figure 2a*). To annotate the 3D masks of the CART19 and K562-CD19 cells, we applied a combination of image processing and the watershed algorithm to a raw RI tomogram according to the following steps (*Figure 2—figure supplement 1*, also see the Codes). First, we annotated the cell masks from a raw RI tomogram using manual selections of four hyper-parameters: (i) initial seed locations of each cell to obtain a 3D distance-transform map, (ii) RI threshold for defining cell boundaries, (iii) voxel dilation sizes for merging over-segmented grains into one discrete region, and (iv) standard deviation of the Gaussian smoothing mask. The processed data were then multiplied to the 3D distance-transform map of the cell regions and segmented by the watershed algorithm. Finally, after iterative adjustment of the parameters from over 2000 data points, three experts in cellular biology quantitatively validated consistent annotation performances via simultaneous truth and performance level estimation (*Figure 2—figure supplement 2*), and heuristically curated the 236 pairs of well-annotated 3D tomograms with consensus (*Warfield et al., 2004*). The curated data uniformly reflected the various stages of the IS dynamics, which ensured providing information about the immunological response in both the early and late stages.

### Training stage

The segmentation tasks were challenged by the lack of distinct boundaries between CART19/K562-CD19 conjugates in RI distributions, diverse morphology of cells, and the demand for precise segmentation at high resolution. As a consequence, although typical segmentation tasks of DCNN were assigned to infer voxel-wise label classification, various types of failure occurred, such as fragmented labels and unnatural IS.

To overcome these limitations and improve the segmentation accuracy and robustness, we designed DCNN to predict the distance map ($y_{pred}$), which was adapted from a previous study (*Wang, 2018*; *Figure 2b*, also see Materials and methods and *Figure 2—figure supplement 3*). We first conducted pre-processing of the 3D annotated masks of CART19 and K562-CD19 cells using the Euclidean distance transform to obtain a true distance map ($y_{true}$). A primary difference from the prior study (*Wang, 2018*) is that the CART19 and K562-CD19 cells were distinguished by the signs of the distance maps (i.e. positive/negative for CART19/K562-CD19 cells). Presently, we set the background to zero.

With the pre-processed data, the DCNN was optimized using the Adam optimizer (Initial learning rate = 0.001, decay rate for the first moment estimate $\beta_1$ = 0.5, decay rate for the second moment estimate $\beta_2$ = 0.99, learning decay rate = 0.5 per 50 epochs) during the training stage to predict the signed 3D distance map of CART19 and K562-CD19 cells. The boundary-weighted L1 function was used as a loss function. To prevent overfitting that might arise from a relatively small number of the training data compared with a large number of parameters, early stopping and data augmentation were used. For early stopping, the obtained annotated dataset was into two disjoint subsets. In one subset, 198 tomogram data pairs were used for the optimization of model parameters. In the other subset, the remaining 36 tomogram data pairs were used for internal validation. Training of the

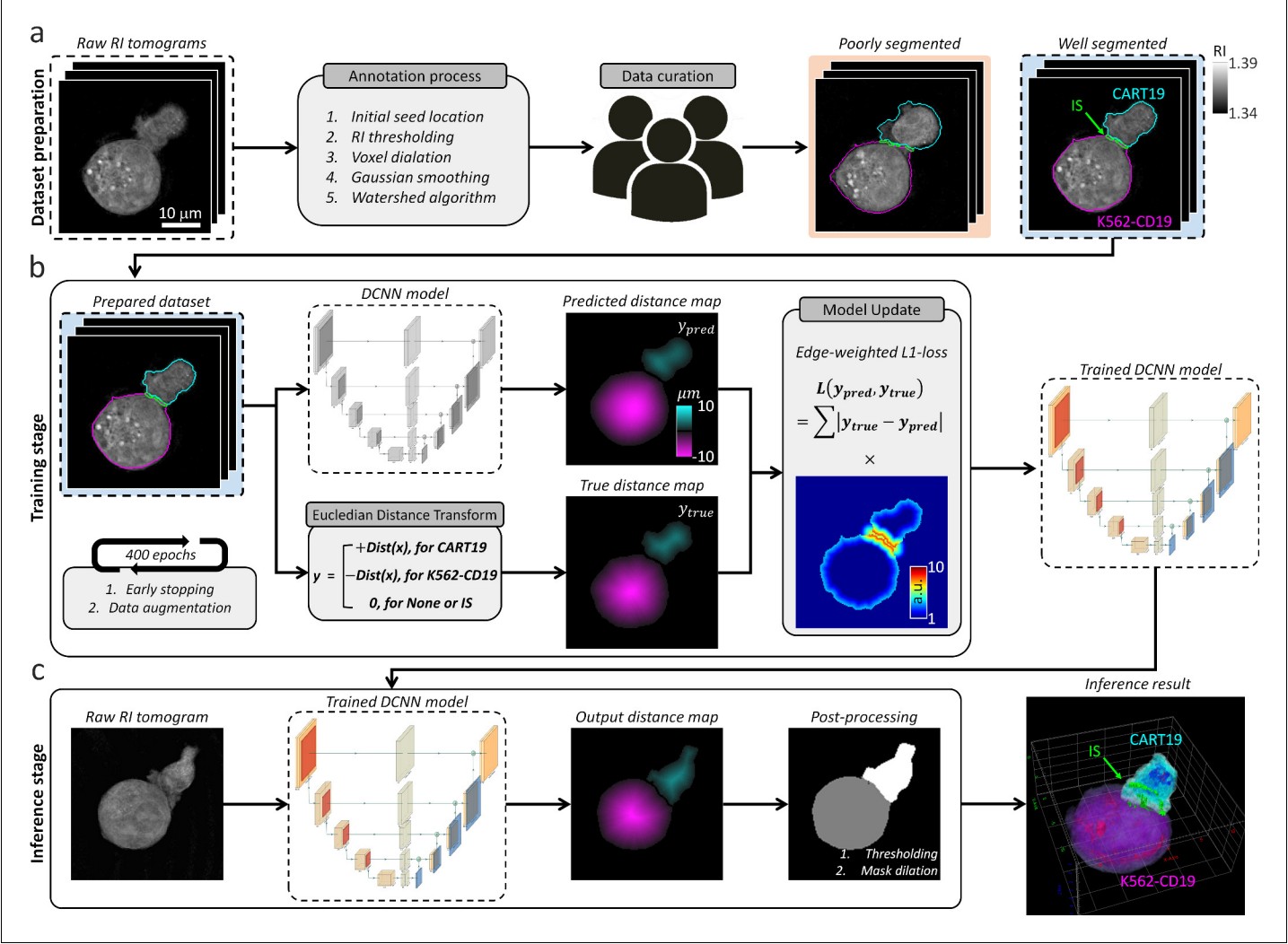

**Figure 2.** Data flowchart in DeepIS framework. (**a**) Dataset preparation; raw RI tomograms were annotated with manual parameter selections and curated by three experts. (**b**) Training stage; the prepared dataset was employed to iteratively train the DCNN model that regressed the distance map of CART19 and K562-CD19 cells with the opposite signs. (**c**) Inference stage; a raw RI tomogram was converted into an output distance map by the trained DCNN model. After post-processing, 3D masks of CART19, K562-CD19, and IS were reconstructed.

The online version of this article includes the following figure supplement(s) for figure 2:

**Figure supplement 1.** The details of the annotation process.

**Figure supplement 2.** Simultaneous truth and performance level estimation between three experts and annotation masks generated by manual parameter selections.

**Figure supplement 3.** The schematic architecture of our deep neural network.

**Figure supplement 4.** Definition of cell interface, interior, and synapse.

model parameters was stopped if the performance of the model on the validation set had not improved for five consecutive epochs. For data augmentation, a larger annotated dataset was simulated using random rotations, horizontal reflection, cropping, and elastic transformation to make the resulting model more robust to irrelevant sources of variability. The network was trained on four graphics processing units (GPUs; GEFORCE GTX 1080 Ti) for 400 epochs, which took approximately 6 hr. Selection of a model for inference among trained models was based on performance on the internal validation set.

## Inference stage

In the inference stage, the trained network was used to regress the distance maps of the CART19 and K562-CD19 cells from unlabeled RI tomograms (*Figure 2c*). In the post-processing stage, the output distance map was post-processed to yield cell domains masks of each CART19 and K562-CD19 cell through simple thresholding using a value of 54.5 nm, which is approximately half of a voxel pitch. The IS of CART19/K562-CD19 conjugate was defined by dilating the CART19 and K562-CD19 masks by a comparable length to the axial resolution of ODT (two voxels, 437 nm) and finding an overlapping region. The cell surface and the cell interior were obtained using binary image erosion by one voxel (218 nm) and defined as the boundary of the 3D cell mask and the remaining interior mask respectively (*Figure 2—figure supplement 4*).

## Evaluation of DeepIS segmentation performance

Because the label-free segmentation approach aims to distinguish the boundaries between two attached cells at a sub-micrometer spatial resolution, it was essential to validate whether DeepIS provides sufficient segmentation accuracy for our purpose. This was first addressed by comparing the segmentation performances between DeepIS and manual segmentation (*Figure 3*, also see *Figure 3—video 1* and Datasets). In the training dataset, this comparison showed that the model could define IS boundaries. Notably, DeepIS generally displayed better segmentation performance than the manual segmentation in the untrained dataset, without notable segmentation problems such as fragmentations and discontinuous boundaries. This observation indicated that DeepIS was well trained and could be exploited to predict the IS boundaries.

We then examined whether DeepIS could be applied for high-throughput analysis of IS morphology. Wide-area segmentation of CART19 and K562-CD19 cells was carried out over a lateral field-

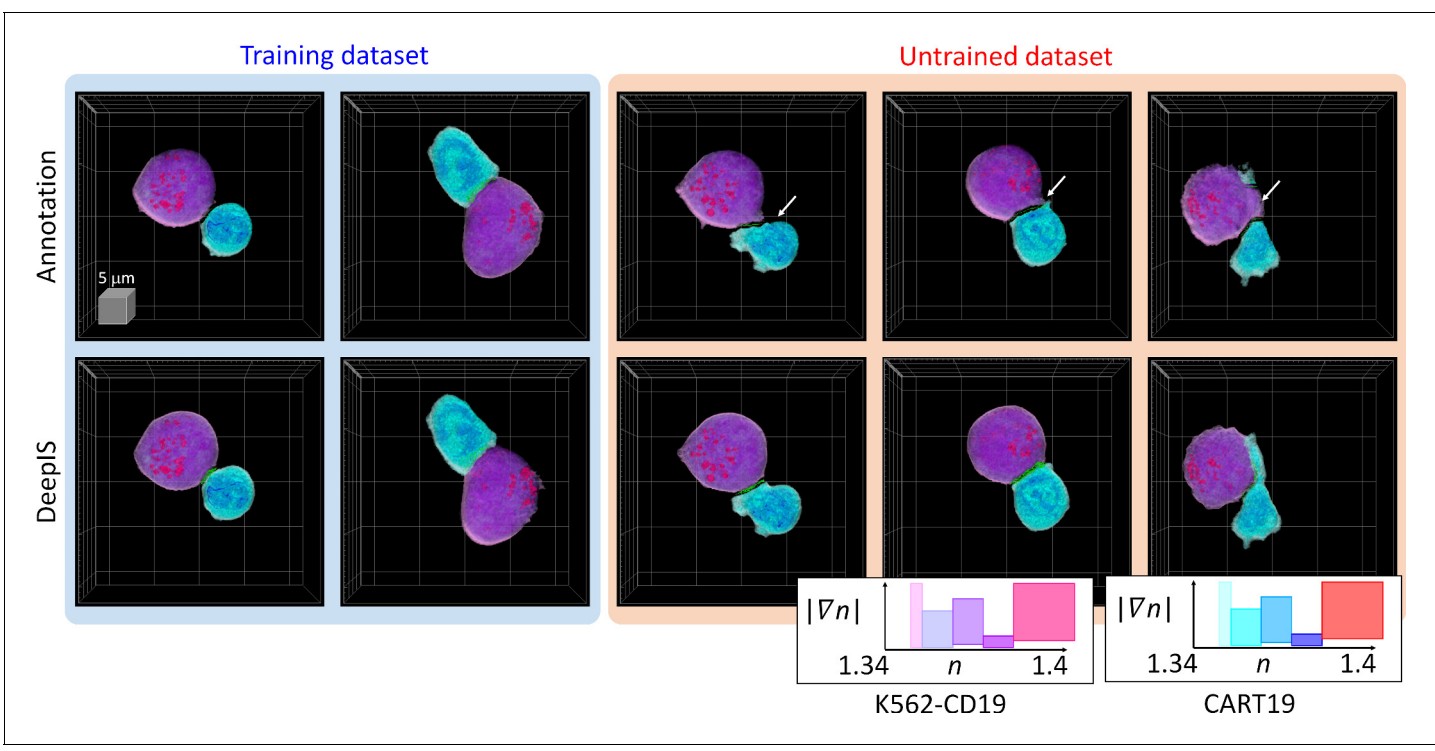

**Figure 3.** Representative segmentation results. (Top row) Masks annotated by manual parameter selections. (Bottom row) Segmentation results using DeepIS. Blue shade indicates the curated data in the training stage. Red shade indicates the data which were poorly segmented by parameter-based annotation. White arrows illustrate poorly segmented regions. Lower right graph: rendering colormaps determined by the ranges of RI and RI gradient norm.

The online version of this article includes the following video for figure 3:

**Figure 3—video 1.** A number of annotated data and DeepIS results for training and untrained datasets.

https://elifesciences.org/articles/49023#fig3video1

of-view exceeding 1 mm² (***Figure 4a***). When cells were located based on RI contrast, DeepIS allowed rapid, automated, and on-site semantic segmentation (***Figure 4b***). Interestingly, DeepIS successfully labeled adjoining CART19/K562-CD19 cell conjugates as well as individual CART19 and K562-CD19 cells (***Figure 4c***), which validated the high-throughput, general segmentation performance of DeepIS.

The segmentation performance of DeepIS was further evaluated by quantifying the segmentation accuracy using manually delineated 3D labels obtained from correlative fluorescence microscopy (***Figure 5***). To prepare 3D manual labels, we mixed T cells and K562 cells expressing CAR-G4S-mCherry and hCD19-G4S-Zsgreen fusion proteins, respectively, and imaged fixed cell conjugates using correlative RI and fluorescence microscopy to delineate manual labels (***Figure 5a***, Materials and methods). We found that RI and fluorescence images overlapped well with the plasma-membrane fluorescence images, confirming that correlative RI and CAR/CD19 fluorescence images could sufficiently resolve the 3D cell topologies (***Figure 5—figure supplement 1***). Furthermore, multi-color 3D confocal microscopy showed with higher spatial resolution that CAR and CD19 were spread throughout the plasma membranes of CART19 and K562-CD19 cells respectively and coalesced into the IS, validating that CAR/CD19 fluorescence images were sufficient for defining the IS (***Figure 5—figure supplement 2***). We then compared the manually drawn labels with the segmentation masks

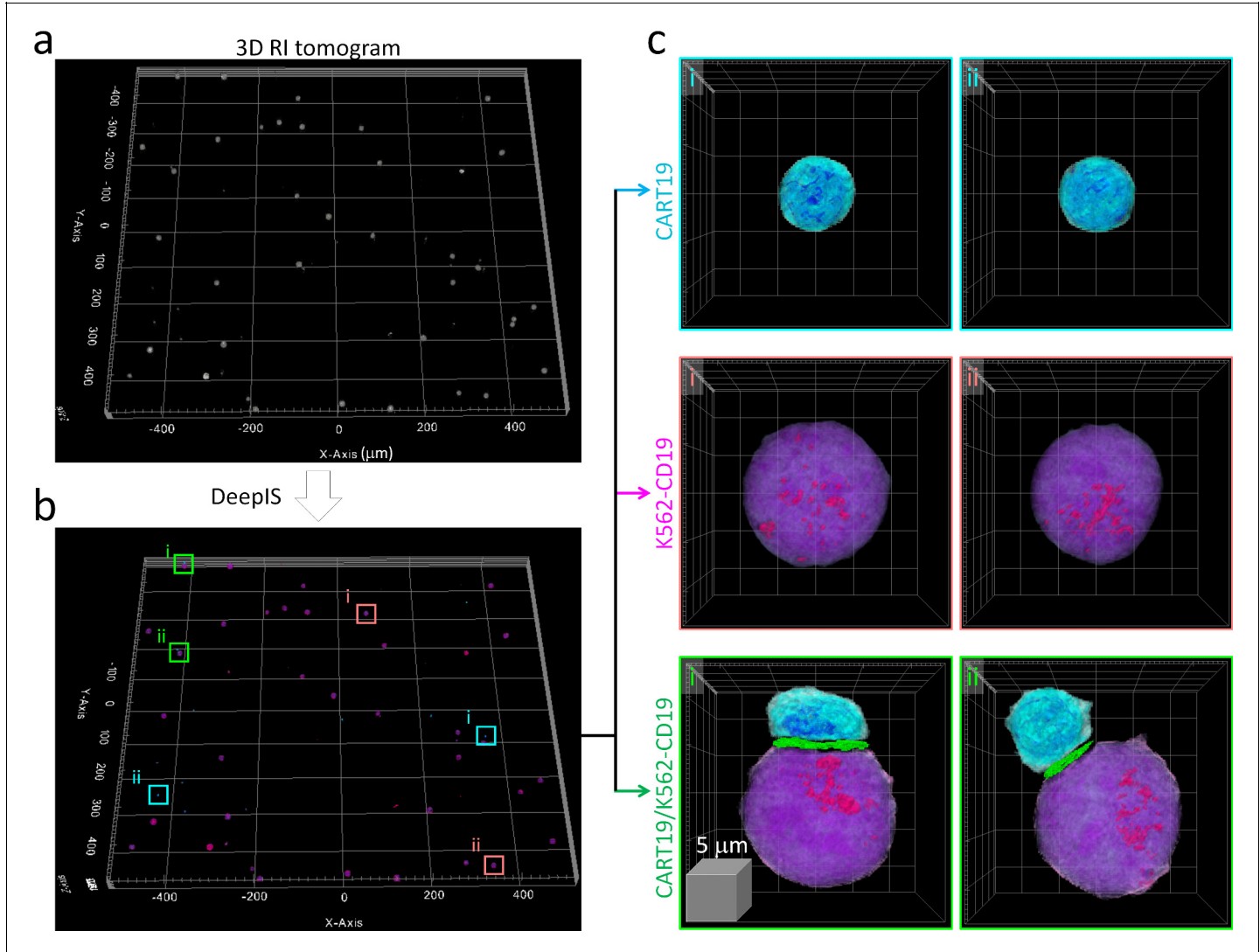

**Figure 4.** High-throughput semantic segmentation using DeepIS. (a) RI tomogram over 0.98 × 1.05 × 0.04 mm³ obtained by stitching method. (b) Segmentation using DeepIS. (c) Representative CART19, K562-CD19, and CART19/K562-CD19 cell conjugates are magnified.

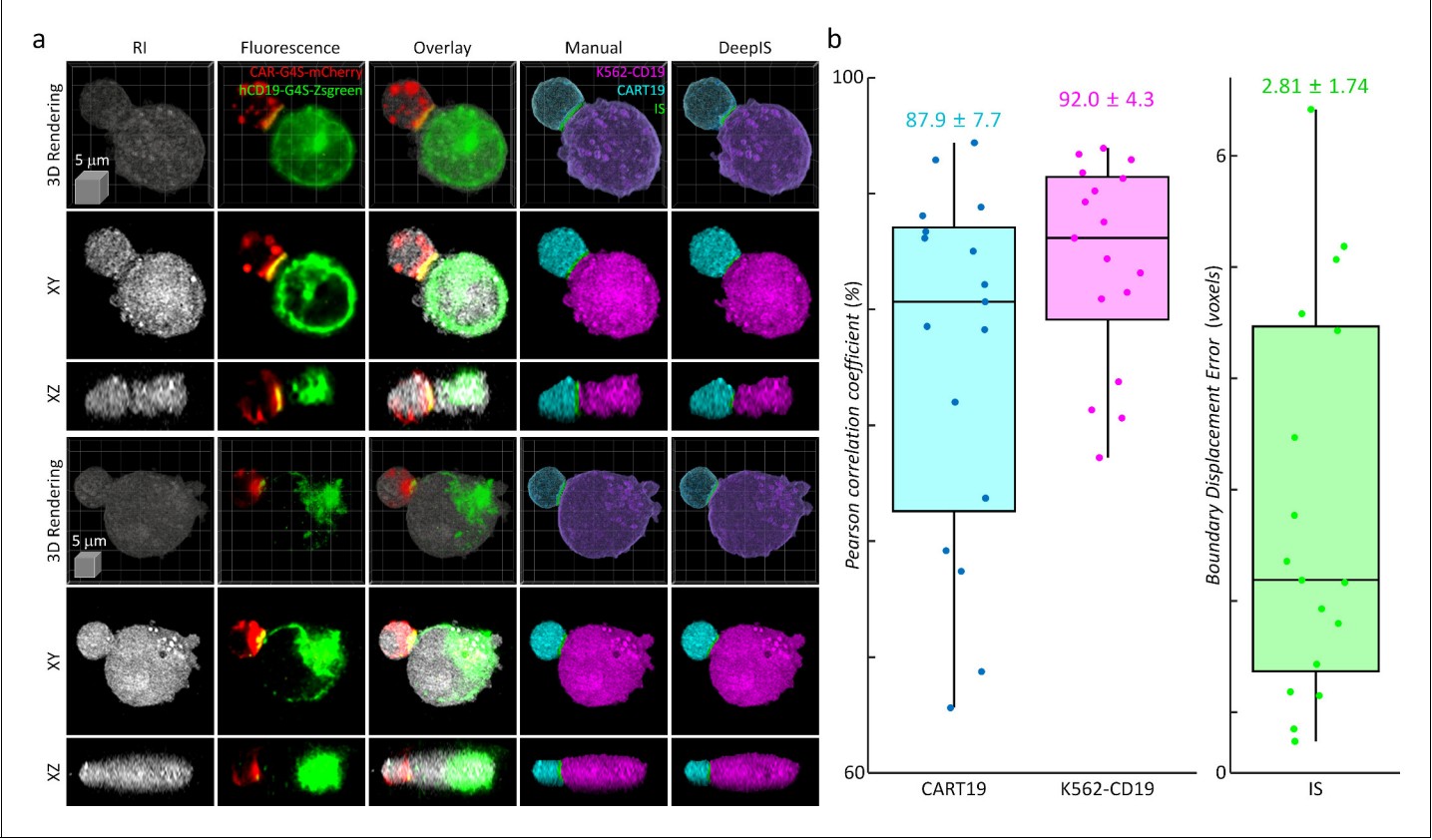

**Figure 5.** Quantitative analysis of segmentation performance using correlative fluorescence microscopy and ODT. (a) Representative 3d rendering images, the *xy*- and the *xz*- cross-sections of RI (first column), fluorescence (second column), and their overlapped images (third column). CART19 and K562-CD19 labels were manually delineated based on the correlative images (fourth column) and compared with the labels obtained from DeepIS (fifth column). (b) Quantifications of DeepIS segmentation performance (*n* = 17). Pearson correlation coefficient was measured for CART19 (87.9 ± 7.7%) and K562-CD19 (92.0 ± 4.3%). Boundary displacement error was measured for IS (2.82 ± 1.74 voxels; 615.5 ± 379.8 nm). Each boxplot indicates the median, upper, and lower quartiles of each population.

The online version of this article includes the following figure supplement(s) for figure 5:

**Figure supplement 1.** Membrane staining indicates good overlaps between RI maps and 3D cell topologies and results in comparable segmentation performance to the experiment without membrane staining.

**Figure supplement 2.** 3D multi-color confocal microscopic images show that membrane CAR and CD19 are concentrated on the immunological synapse and generated at the plasma membrane between CART19 and K562-CD19 cells.

obtained from DeepIS (*Figure 5b*). When 3D Pearson correlation coefficients for volumetric masks were quantified, we obtained the mean ± standard deviation (SD) values of 87.9 ± 7.7% and 92.0 ± 4.3% for CART19 and K562-CD19 cells, respectively, implying a greater than 80% good overlap between the manual labels and automatically segmented labels using DeepIS. In addition, the mean ± SD value for boundary displacement error was 2.82 ± 1.74 voxels, which corresponded to a sub-micrometer displacement error of 615.5 ± 379.8 nm. Considering that 3D segmentation is more challenging than 2D segmentation, these results indicated significantly small boundary errors (*Wang, 2018*; *Lalaoui and Mohamadi, 2013*).

For a conclusive validation of the segmentation performance, we lastly compared the IS defined by DeepIS with the IS imaged by high-resolution fluorescence microscopy methods. For this purpose, we integrated the DeepIS framework with multicolor 3D structured illumination microscopy (3D-SIM; see Materials and methods) (*Mobahi et al., 2011*; *Gustafsson et al., 2008*; *Demmerle et al., 2017*; *Kim et al., 2017*). The integrated setup enabled us to image the protein compositions at CAR IS at sub-200-nm lateral and nearly sub-400-nm axial spatial resolution defined as full width at half-maximum (*Figure 6—figure supplement 1*).

Using 3D-SIM, we first assessed the 3D location of the synaptic cleft (*Figure 6*). We used the negative stain method by adding the FITC-labeled dextran solution with two different hydrodynamic diameters (4 and 54 nm) into the chemically fixed conjugates of K562-CD19 cells and T cells expressing CAR-G4S-mCherry (see Materials and methods). The resultant 3D-SIM images showed that the

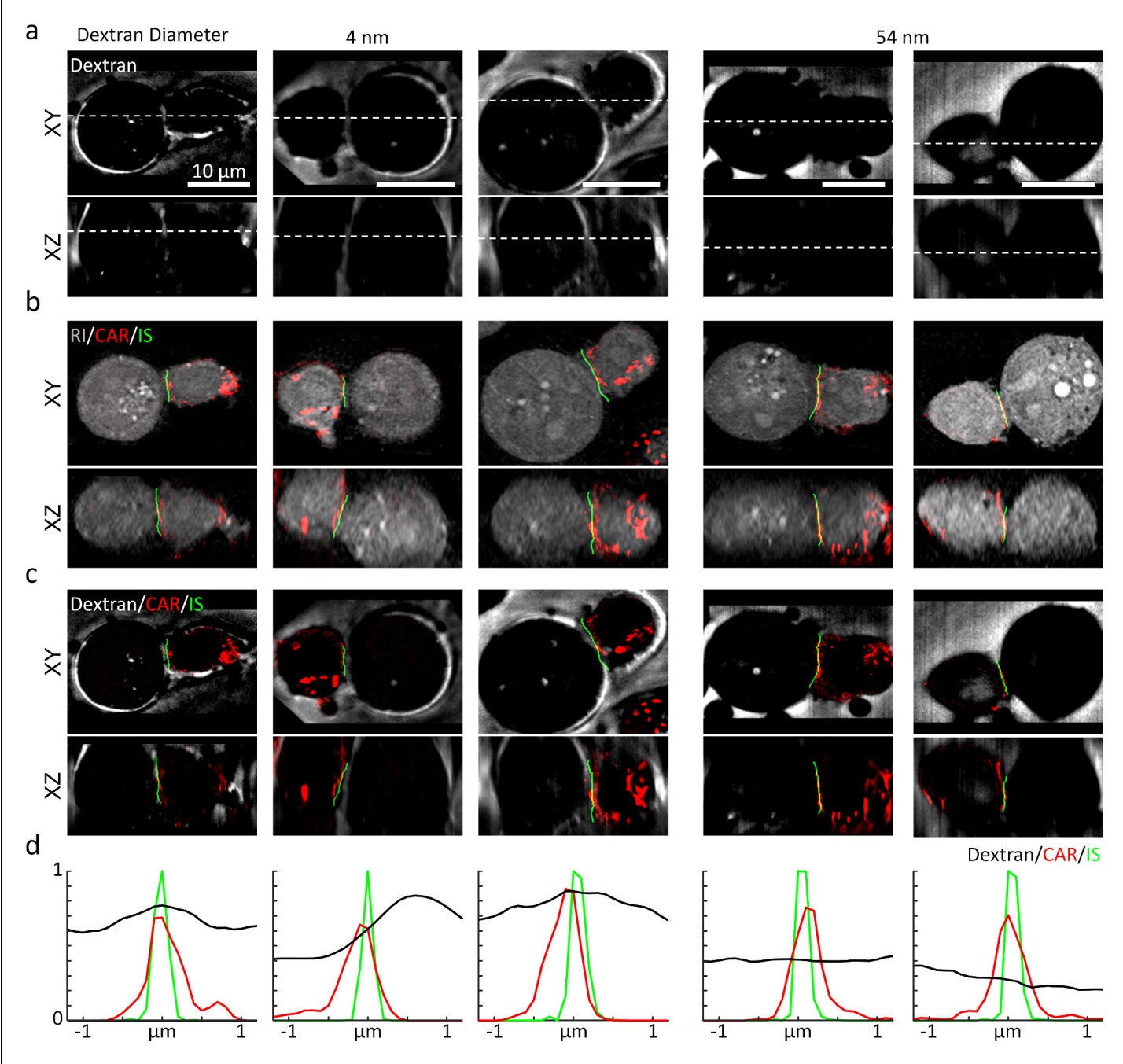

**Figure 6.** High-resolution validation of 3D IS using a negative stain method. (**a**) The XY- and XZ-slice images of fluorescein-labeled dextran. White dashed lines: slice regions. (**b**) Corresponding XY- and XZ- images of SIM and ODT. (**c**) Comparison of dextran fluorescence, CAR fluorescence, and the IS defined by DeepIS. (**d**) Mean lateral line profiles for lines drawn across the IS.

The online version of this article includes the following figure supplement(s) for figure 6:

**Figure supplement 1.** Implementations of complementary 3D-SIM and ODT.

**Figure supplement 2.** CAR IS excludes dextran molecules in a size-dependent manner.

synaptic cleft was visible only when we used the smaller dextran (4 nm), suggesting that, similar with the IS of NK cells, CAR IS excluded dextran molecules above a size threshold (*Cartwright et al., 2014*; *Figure 6a*, also see *Figure 6—figure supplement 2*). We then compared the imaged synaptic clefts, the CAR fluorescence, and the IS drawn by DeepIS for validation (*Figure 6b*). The 3D overlaps of the three images validated the 3D segmentation accuracy of our proposed method (*Figure 6c*). We laterally plotted the signals across the IS to quantitatively assess the segmentation accuracy (*Figure 6d*). The displacements of the IS from the peak intensities of the CAR and the synaptic cleft were within 200 and 600 nm, respectively. The result suggested that the IS drawn by DeepIS reflected the IS boundary closer to CAR-T cells, whose accuracy was comparable to the resolution limit of ODT. Moreover, consistent with the previous study using confocal fluorescence microscopy (*Cartwright et al., 2014*), the lateral full-width half-maximum of the synaptic cleft was measured to be 894 nm, which may imply the presence of a spatial gap larger than tens of nanometers across the IS (*McCann et al., 2003*).

Additionally, we validated whether the proposed method could be used to detect organelles at CAR IS. We specifically investigated the distributions of lytic granules, CAR and CD19 proteins, by imaging T cells expressing CD19-specific CAR-G4S-mCherry stained with lysotracker dyes and K562 cells expressing hCD19-G4S-Zsgreen. To verify successful staining of lytic granules before 3D-SIM, we employed widefield deconvolution fluorescence microscopy and confirmed the rapid dynamics of lytic granules coalescing into the IS (*Figure 7—figure supplement 1*, also see *Figure 7—video 1*). We also confirmed the RI detection sensitivity of the lytic granules using a histogram analysis, which showed that, compared to cell bodies, the mean RI value was lower for CAR IS and higher for lytic granules (*Figure 7—figure supplement 2*). We then chemically fixed stable CART19/K562-CD19 conjugates and imaged them using 3D-SIM and ODT for high-resolution multi-protein composition analysis (see Materials and methods).

As a result, our DeepIS framework demarcated the interface regions in close proximity to the overlapping areas of CAR and CD19 proteins (*Figure 7a*). Importantly, the demarcated region by DeepIS was consistent with the 3D-SIM images, which provided clearer 3D features of the IS between CART19 and K562-CD19 cells than widefield fluorescence microscopy. Along the IS demarcated by DeepIS, we quantitatively analyzed the *en face* images of CAR, CD19 and lytic granules imaged by 3D-SIM (*Figure 7b*). In agreement with the previous report (*Davenport et al., 2018*), the protein compositions of CAR exhibited asymmetric and granular distributions along the CAR IS. We analyzed the correlations between the total protein concentration distribution and the imaged proteins at CAR IS. The correlative line and surface profiles of the protein signals indicated the highest correlations of CAR with lytic granules, as well as colocalizations of CD19 proteins with the CAR clusters (*Figure 7c*). Interestingly, the total surficial protein densities approximated by ODT exhibited both correlated and uncorrelated clustered regions with the dense multi-protein clusters. Since ODT quantitatively estimates the total protein concentration, the uncorrelated signals are highly likely to imply the presence of clusters of other dominant proteins such as F-actin, Lck, and supramolecular attack particles (*Xiong et al., 2018*; *Bálint et al., 2020*).

Taken together, these results suggest that our DeepIS method based on ODT can be used to define IS area with high accuracy, and also provides collective information about the distribution of total proteins within the IS which may not be easily measured by using conventional high-resolution fluorescence microscopy.

## Quantitative kinetic analysis of CART19 IS formation using DeepIS

The successfully established DeepIS was implemented in the detailed kinetic analysis of the IS formation between CART19 and K562-CD19 cells using morphological and biochemical parameters (*Figure 8*). Specifically, we analyzed 27 sets of IS dynamic datasets measured over 300 s to 10 min at time intervals of 3 to 8 s to determine the kinetics of synapse area, membrane protein density, and intracellular protein density.

First, we examined the temporal changes of synapse areas depending on the expression of the target antigen, CD19, on K562 cells (*Figure 8a*). As expected, CART19 cells could not form a stable synapse with K562 cells (CD19-negative) in five independent experimental trials (*Figure 8a*, also see *Figure 8—video 1*). By contrast, CART19 cells formed a stable IS with K562-CD19 cells (CD19-positive), and induced apoptotic blebbing on the target cells about 9 min after the initial contact (*Figure 8—video 2*). For the statistical analysis of the initial IS area changes, DeepIS was applied to both

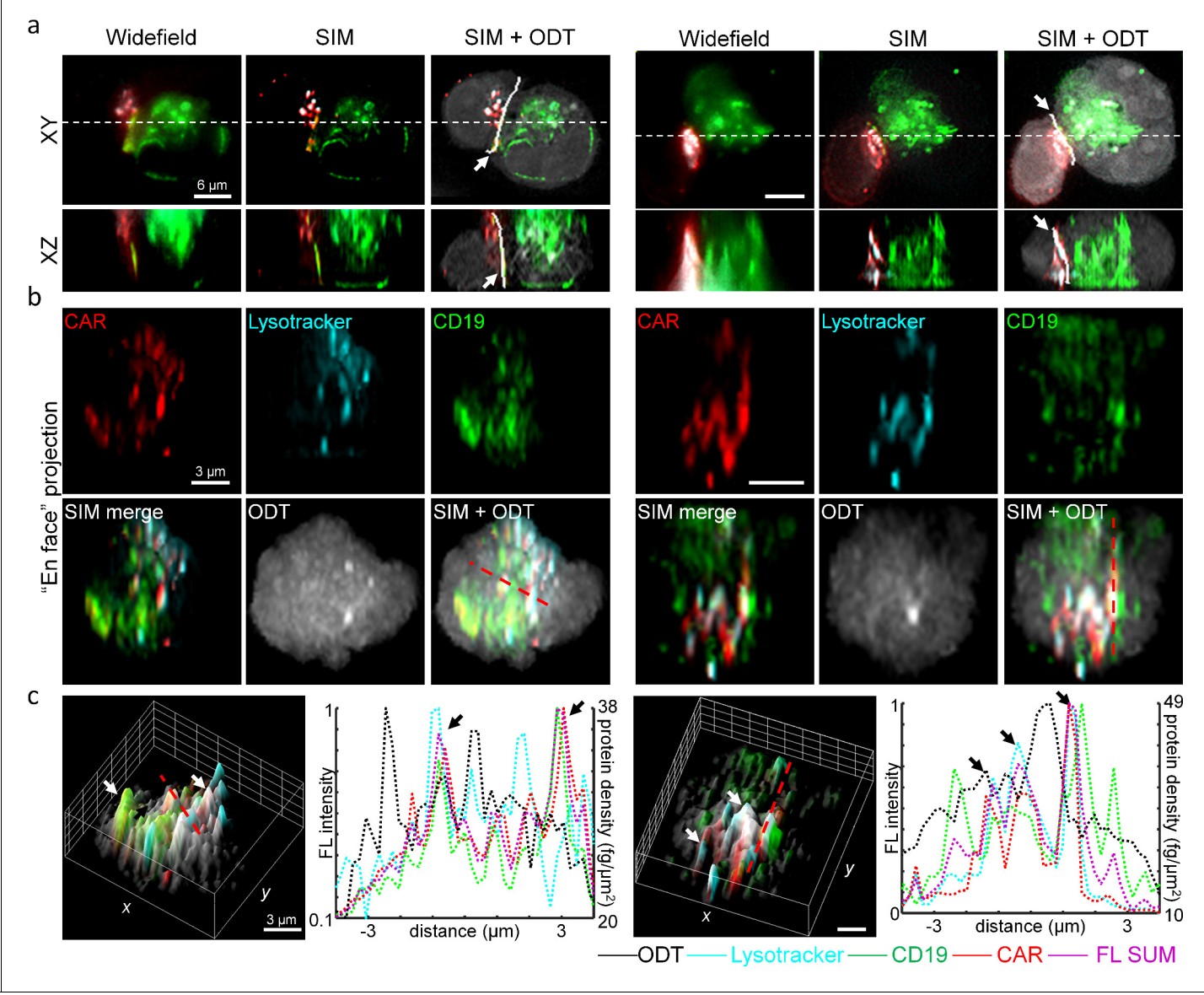

**Figure 7.** High-resolution analysis of 3D IS compositions using 3D-SIM and DeepIS. (**a**) XY- and XZ-slice images of representative CART19/K562-CD19 conjugates. White-dashed lines: slice regions. White bold lines in SIM + ODT: 3D IS areas defined by DeepIS. (**b**) *En face* projected images of the defined IS areas in a. (**c**) Surface (left each) and line (right each) profiles along the dashed lines of normalized fluorescence intensities and surficial protein densities. Arrows indicate representative colocalized signals.

The online version of this article includes the following video and figure supplement(s) for figure 7:

**Figure supplement 1.** 3D lytic granule transport imaged by ODT and 3D widefield deconvolution microscopy.

**Figure supplement 2.** RI sensitivity analysis.

**Figure 7—video 1.** Two 3D dynamic videos of ODT and 3D widefield deconvolution fluorescence imaging showing rapid coalesce of lytic granules to the IS.

https://elifesciences.org/articles/49023#fig7video1

dynamic datasets and successfully segmented CART19 cells, target cells, and IS boundaries. The individual temporal plots of IS showed that each data exhibited smooth IS increase and indicated no undulation problems due to ill segmentations in most cases (*Figure 8—figure supplement 1*). The temporal graphs of the mean synapse area showed that, whereas the IS for K562 cells was not stably formed for 300 s, the IS for K562-CD19 expanded to the half of the maximum synapse area ($A_{max}$ = 106.16 µm²) within 40 s and reached a steady-state within only 3 min (*Figure 8b*).

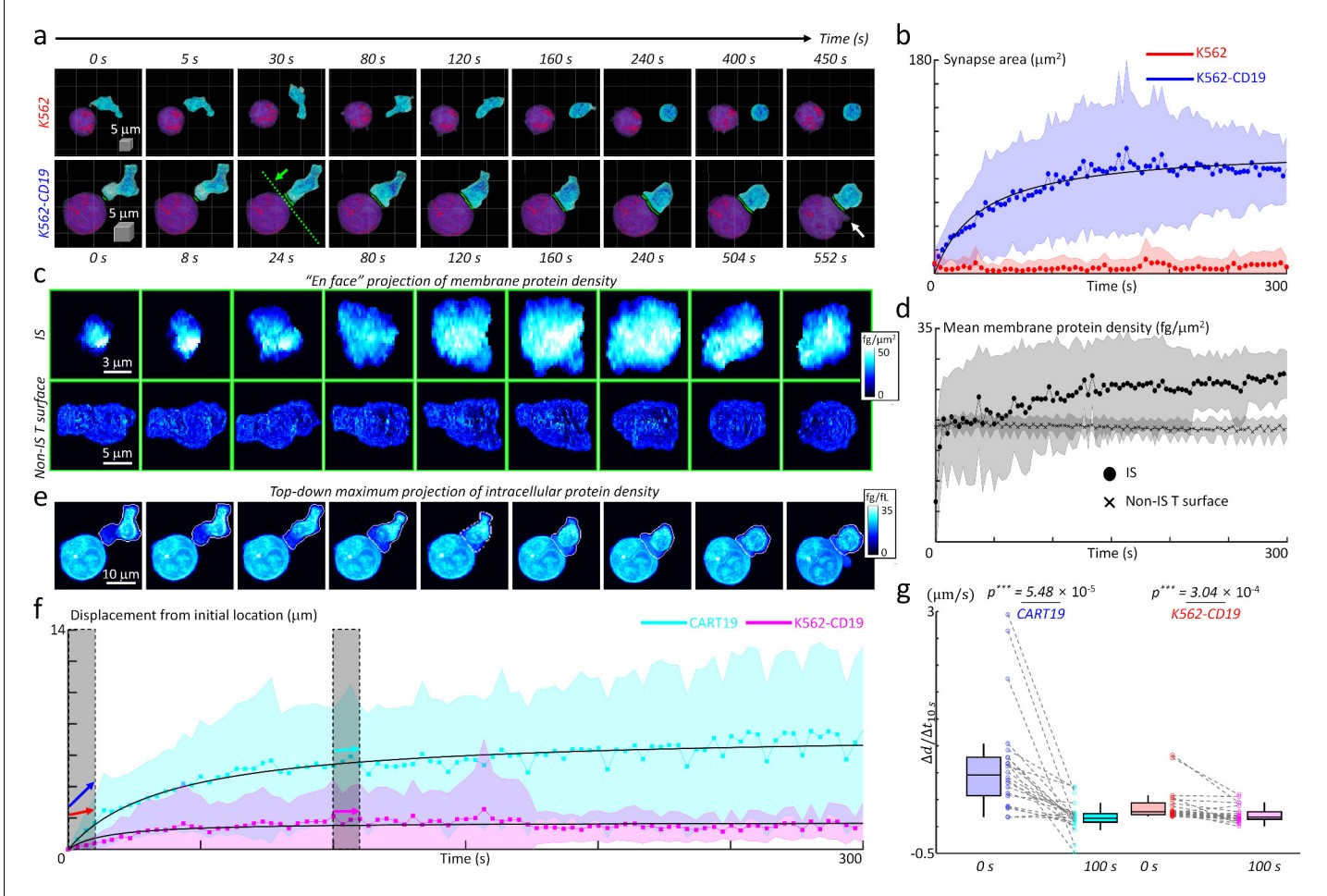

**Figure 8.** Quantification of initial IS formation kinetics of CART19 cells. (**a**) Representative snapshots of a video of CART19 cells responding to K562 (top row) and K562-CD19 (bottom row) cells. Purple: K562, Blue: CART19. 0 s: the initial contact point of the effector cell and target cells. White arrow: the blebbing point. (**b**) Temporal changes in the synapse areas of CART19 cells responding to K562 cells ($n = 5$) and K562-CD19 cells ($n = 22$). Black line: fitting curve with $A(t)=A_{max}t/(t + \tau_{1/2})$, where $A_{max} = 106.16$ μm$^2$ and $\tau_{1/2} = 39.63$ seconds, Pearson correlation coefficient ($\rho$)=0.93. (**c**) Maximum *en face* projection of membrane protein density of IS (top row) and non-IS T surface (bottom row) for CART19/K562-CD19 conjugate. The direction of the en face projection and the projection plane: the green arrow and a dashed line in (**a**). (**d**) Temporal changes in the mean membrane protein density of CART19 immunological synapses (circles) and non-IS T surfaces (crosses) responding to K562-CD19 cells. (**e**) Maximum *z*-axis projection of intracellular protein density distributions of the CART19/K562-CD19 conjugate. The white contours: the boundaries of the CART19 mask. (**f**) Temporal changes in the displacements of the center-of-masses of CART19 and K562-CD19 cells from their initial locations. Black lines: $\Delta d(t) = \Delta d_{max} t/(t + \tau_{1/2})$, where $\Delta d_{max} = 7.48$ and $1.74$ μm, $\tau_{1/2} = 37.84$ and $14.56$ s, $\rho = 0.91$ and $0.58$ for CART19 and K562-CD19 respectively. (**g**) The average changes for 10 s in the early (0 s) and late (100 s) stages are marked by colored arrows, and statistically compared by two-tail paired Wilcoxon tests for CART19 and K562-CD19 cells, respectively. CART19: $\Delta d/\Delta t_{10s} = 818.5 \pm 788.5$ nm/s, and $26.6 \pm 223.6$ nm/s in the early and late stages. K562-CD19: $\Delta d/\Delta t_{10s} = 194.3 \pm 246.5$ nm/s, and $63.7 \pm 125.6$ nm/s in the early and late stages. *** indicates p<0.001. Each shaded error bar indicates the mean and standard deviation of line plots. Each boxplot indicates the median, upper, and lower quartiles of each population. The central points and shades in *Figures 7b, d and f* indicate the mean and standard deviations respectively. The numbers of analyzed cells per each experiment are listed in *Table 1*.

The online version of this article includes the following video and figure supplement(s) for figure 8:

**Figure supplement 1.** Individual temporal synapse area graphs of CART19 cells responding to K562-CD19 cells ($n = 22$).

**Figure 8—video 1.** A CART19 cell fails to form a stable IS with a K562 cell.

https://elifesciences.org/articles/49023#fig8video1

**Figure 8—video 2.** CART19 cells formed a stable IS with K562-CD19 cells and induced apoptotic blebbing on the target cells about 9 min after the initial contact.

https://elifesciences.org/articles/49023#fig8video2

**Figure 8—video 3.** A CART19 cell exerted mechanical forces during the dynamic IS formation that led to the subsequent translational displacement and deformation of the K562-CD19 cells.

https://elifesciences.org/articles/49023#fig8video3

Next, we also assessed whether the membrane protein amount differs between the IS and non-IS areas of CART19 cells by comparing the 2D maximum projected snapshots of membrane protein densities in each area (*Figure 8c*). Within the IS surface, a dramatic increase in membrane protein density, as well as the synapse area, was observed. By contrast, the non-IS T surfaces of CART19 cells maintained the constant, lower membrane protein densities. In line with this observation, quantification of temporal changes in the mean membrane protein densities revealed that up to $27 \pm 4$ fg/$\mu m^2$ at 300 s have been accumulated in the IS surface, which was higher than average protein density ($19 \pm 1$ fg/$\mu m^2$) in the non-IS surface (*Figure 8d*).

We further explored the cell mechanics of CART19 and K562-CD19 cells during their initial IS formation. As in *Figure 8e*, the time-lapse snapshots of maximum 2D projection of intracellular protein densities visualized the dynamic action of CART19 cells, which incorporated the polarization of the intracellular protein contents. Furthermore, CART19 cells exerted mechanical forces during the dynamic IS formation, which led to the subsequent translational displacement of the K562-CD19 cells (*Figure 8—video 3*). To quantify the cell translocation dynamics, we traced the temporal changes in the displacement of the center-of-mass (the average displacement weighted by the intracellular protein density) with respect to the initial location for each cell (*Figure 8f*). As observed, CART19 cells exhibited more dynamic motions ($\Delta d_{max} = 7.48$ $\mu m$) than K562-CD19 cells ($\Delta d_{max} = 1.74$ $\mu m$) during the IS formation. For both cells, the mechanical translocations were more dramatic in the earlier stage of the IS formation, as confirmed statistically by comparing the average change in the cell translocations in the early- and late-stage (*Figure 8g*).

Overall, the results presented here generally recapitulate previously observed phenomena in the synapse studies based on conventional microscopy techniques. The rapid increase in the membrane protein density in the synapse area likely reflects the influx of large amounts of IS protein components, including CARs, actins, and other adhesion molecules (*Xiong et al., 2018*). In addition, perhaps the intracellular protein density changes in CART19 cells, which indicates polarization of the intracellular organelles in CART19 cells until stabilization, can be explained by the centrosome polarization of cytotoxic T lymphocytes (*Ritter et al., 2015*). Lastly, the force exerted by the CART19 cells on target cells during the IS formation suggests that CART19 cells also integrate mechanical potentiation during target cell killing similar to TCR-based cytotoxic T cells (*Basu et al., 2016*).

## Statistical analysis of IS parameters depending on the co-stimulatory domains of CAR

Kymriah and Yescarta are the only two CD19-targeting CAR-T cell therapies that have been approved by US-FDA to date. Since one of the major differences between the two 2[nd] generation CAR-T cells lies in the sequence of the costimulatory signaling domain used (4-1BB for Kymriah and CD28 for Yescarta), we next thought to apply our method to compare the IS characteristics of the CAR-T cells with different costimulatory signaling domains. Primary human T cells transduced with the lentiviral vectors encoding CD19-28z or CD19-BBz CAR showed comparable transduction efficiency as well as similar surface expression levels of CARs as determined by flow cytometry (*Figure 9—figure supplement 1*).

Exploiting our established method, we studied the effect of CD28 or 4-1BB co-stimulatory domains on CAR IS characteristics (see Materials and methods). We first observed the early IS dynamics between CD28 and 4-1BB based CAR-T cells in the presence of K562-CD19 target cells, and found the stable IS formation within five minutes without statistically significant kinetic differences (*Figure 9—video 1*, also see *Figure 9—figure supplement 2*). We then compared the statistics of other IS parameters between the CD19-28z and CD19-BBz CAR-T cells in a steady state. Specifically, we incubated each type of CAR-T cells with K562-CD19 cells for 15 min to allow sufficient time for stable IS formation, and fixed them with 4% paraformaldehyde solution. When we analyzed the images for conjugates (*Figure 9a*), we found no significant difference in IS areas between the two CAR-T cell types (*Figure 9b*). However, statistical analysis indicated significantly higher IS protein densities and total IS protein amounts for CD19-BBz CAR-T cells, approximately by 10% compared with CD19-28z CAR-T cells (*Figure 9c–d*). Collectively, our results indicate that the quantitative analysis of IS parameters using DeepIS, in conjunction with other analytical methods such as fluorescence-based microscopy and quantitative mass-spectrometry (*Salter et al., 2018*), may help to elucidate the mechanistic details underlying the functional differences observed for the CAR-T cells

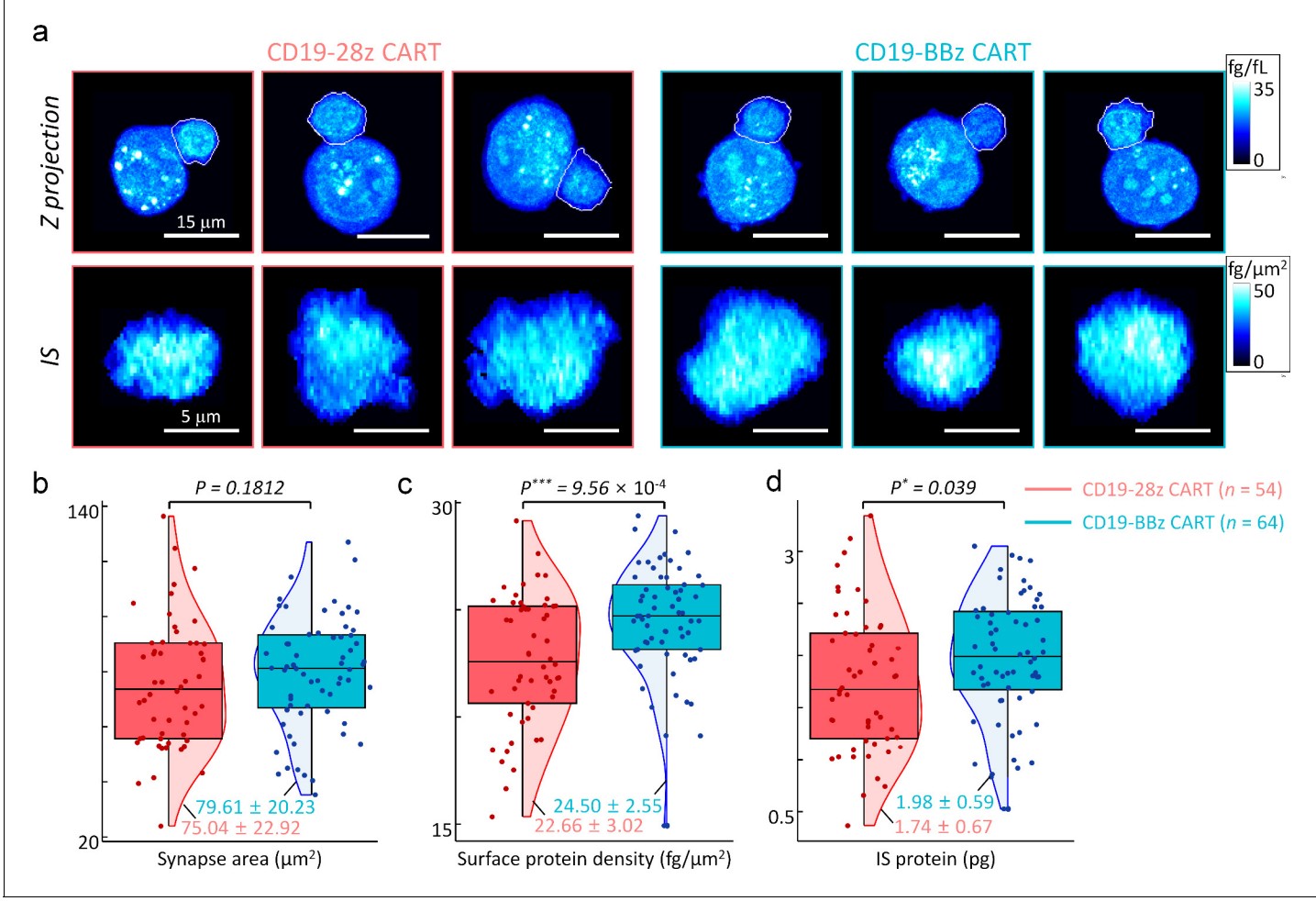

**Figure 9.** Statistical analyses of synapse morphologies depending on the co-stimulatory domains. (a) Representative images for maximum *z*-axis projection of intracellular protein density distributions (top row) and *en face* projection of surface protein density distributions of IS (bottom row). White boundaries indicate the annotated membranes of CART19 cells using DeepIS. (b–d) Scatterplots of (b) synapse areas, (c) mean surface protein density, and (d) total IS protein amount. Each boxplot indicates the median, upper, and lower quartiles of each population. The vertical lines indicate population ranges. Perpendicular shades indicate normalized population density distributions. Two-tail unpaired Wilcoxon tests were performed. mean ± SD data are presented. The numbers of analyzed cells per each experiment are listed in *Table 1*.
The online version of this article includes the following video and figure supplement(s) for figure 9:

**Figure supplement 1.** Generations of CD19 specific CAR-T cells and hCD19 transduced target cells for investigating synapse formation dynamics.
**Figure supplement 2.** Individual temporal synapse area curves of CD19-28z (*n* = 3) and CD19-BBz (*n* = 5) CART19 cells.
**Figure 9—video 1.** Both CD28 and 4-1BB based CART19 cells formed stable IS within several minutes.
https://elifesciences.org/articles/49023#fig9video1

with different signaling domains (*Lee and Kim, 2019*; *van der Stegen et al., 2015*; *Guedan et al., 2019*).

## Discussion

The collective results demonstrate that DeepIS in combination with ODT and deep neural network enables the label-free, time-lapse 3D IS tracking in an automated manner. The successful segmentation performance of DeepIS could not be possible without careful validations in the initial development stage, including data curation, model training, and qualitative and quantitative tests for general segmentation capabilities. This platform was applied to define the IS parameters related to their morphological and biochemical traits, and to quantify the IS dynamics of CART19 cells. We anticipate that the proposed method can be generalized to study a broad range of IS that are

**Table 1.** Numbers of analyzed cells per each experiment.

| Data | Experiment | 1 | 2 | 3 | 4 | 5 | 6 | 7 | 8 | 9 | 10 | Total |
|------|------------|---|---|---|---|---|---|---|---|---|----|-------|
| # of data | Statistics (*Figure 5*) | 7 | 10 | - | - | - | - | - | - | - | - | 17 |
| | dynamics (*Figure 7*) | 9 | 3 | 3 | 1 | 3 | 3 | 1 | 1 | 2 | 1 | 27 |
| | 4-1BB (*Figure 8*) | 20 | 19 | 25 | - | - | - | - | - | - | - | 64 |
| | CD28 (*Figure 8*) | 1 | 11 | 10 | 9 | 23 | - | - | - | - | - | 54 |
| | Statistics (*Figure 5—figure supplement 1*) | 8 | 22 | - | - | - | - | - | - | - | - | 30 |
| | 10 kDa (*Figure 6—figure supplement 2*) | 3 | 7 | - | - | - | - | - | - | - | - | 10 |
| | 2000 kDa (*Figure 6—figure supplement 2*) | 6 | 3 | - | - | - | - | - | - | - | - | 9 |
| | 4-1BB (*Figure 8—figure supplement 1*) | 1 | 1 | 2 | 1 | - | - | - | - | - | - | 5 |
| | CD28 (*Figure 8—figure supplement 1*) | 2 | 1 | - | - | - | - | - | - | - | - | 3 |

mediated by different types of immune receptors such as TCR and B-cell receptor (BCR) as well as invariant receptors expressed on innate natural killer (NK) cells. In particular, the model will be a powerful way to test whether the IS morphology is affected by chemical treatment and genetic mutations (*Tamzalit et al., 2019*).

Previously, Xiong W et al. have evaluated the quality of the CAR IS formed by CD28 co-stimulatory domain including 2nd CAR (CD19-28z CAR) or CD28 + 4-1BB costimulatory domain including 3rd CAR (CD19-28BBz CAR) through confocal microscopic analysis using glass-supported planar lipid bilayers (*Xiong et al., 2018*). Although they showed that the 3rd generation CAR incorporates more IS-related proteins (F-actin, phospho-CD3z) than 2nd generation CAR, there were no direct comparative experiments on the properties of 2nd CAR IS depending on the types of co-stimulatory domain. Our present work fills the missing gaps, suggesting that CAR IS can also vary among the groups of 2nd generation CAR. In particular, we demonstrated that the surface protein density and total protein amount differ between CD28 and 4-1BB based CAR IS, which recalls the importance of quantitative biochemical analysis of CAR IS.

The present work is, to the best of our knowledge, the first application of ODT and deep learning to understand IS dynamics. Refinements are anticipated in an immediate follow-up study. For example, more rapid immune dynamics can be investigated at higher spatiotemporal resolution in a more sophisticated experimental setup, which may reveal dynamic diffusions of subcellular organelles (*Kim et al., 2016b*) during the immune responses. Also, more efforts in data curation and network design may allow robust time-lapse IS tracking. In order to do so, segmentation based on active contour methods, and new network models based on recurrent neural networks (*Mikolov et al., 2010*), Bayesian neural network (*Snoek, 2015*), and pyramid pooling (*Bian, 2018*) may be incorporated. Although we have demonstrated the label-free imaging and analyzing IS, one of the limitations of the current work is that the error of synaptic cleft determination is on the order of 500 nm. To overcome this, the negative staining with the smallest dextran or correlative electron micrographs may be used to provide a ground truth for improving the precision of synaptic cleft determination.

Our study focused on the platform capable of quantitatively tracking 3D IS dynamics in a completely label-free manner. The RI contrast used in the present study alone does not provide information about individual proteins that are transported during the IS formation of CART19 cells. To provide such biochemically relevant information, we expect that correlative imaging with fluorescence microscopy will circumvent the inherent lack of chemical specificities (*Demmerle et al., 2017*; *Kim et al., 2017*). It also remains unclear how much force was associated during the IS formation, and whether this force correlates with the cytotoxic intensity of CART19 cells. To test the hypothesis, the simultaneous use of microscopic force measurement techniques and ODT will be helpful. For

instance, a combined technique of ODT and holographic optical tweezers (*Kim et al., 2015*; *Kim and Park, 2017*) or traction force microscopy can supplement our DeepIS framework for understanding the mechanical potentiation during the IS kinetics. We expect that these future studies may be exploited for label-free predictions of dynamic molecular transports occurring in the IS formation, and provide complementary information required to elucidate the IS formation mechanisms.

# Materials and methods

## Key resources table

| Reagent type (species) or resource | Designation | Source or reference | Identifiers | Additional information |
|---|---|---|---|---|
| Cell line (*Homo sapiens*) | K562 | Other | RRID:CVCL_0004 | Obtained from Dr. Travis S. Young at California Institute for Biomedical research. The original stock was purchased from ATCC |
| Cell line (*Homo sapiens*) | K562-CD19 | Other | | Obtained from Dr. Travis S. Young at California Institute for Biomedical research. CD19-negative K562 cells were transduced wtih lentiviral vector encoding human CD19. |
| Cell line (*Homo sapiens*) | Lenti-X 293T | Takara | Cat. #: 632180 | Lentivirus production |
| Recombinant DNA reagent | pLV-ΔLNGFR-P2A-CD19-BBz CAR | This paper | | Signal peptide: aa 1–28 hLNGFR (Uniprot TNR16_Human) Extracellular domain: aa 29–250 hLNGFR (Uniprot TNR16_Human) Transmembrane domain: aa 251–272 hLNGFR (Uniprot TNR16_Human) Cytosolic sequence: aa 273–275 (Uniprot TNR16_Human) ΔLNGFR was fused with P2A sequence (GGAAGCGGAGCTACTAACTTCAGCCTGCTGAAGCAGGCTGGCGACGTGGAGGAGAACCCTGGACCT) and followed by CAR sequence Signal peptide: aa 2–21 hCD8 (Uniprot CD8A_Human) Extracellular domain: anti-CD19 scFv (Clone: FMC 63); VL –G4S linker VH Hinge/Transmembrane domain aa 138–206 hCD8 (Uniprot CD8A_Human) Co-stimulatory domain sequence: aa 214–255 (Uniprot TNR9_Human) Signaling domain: aa 52–164 (Uniprot CD3Z_Human) |
| Recombinant DNA reagent | pLV- ΔLNGFR -P2A-CD19-28z CAR | This paper | | Signal peptide: aa 1–28 (Uniprot TNR16_Human) Extracellular domain: aa 29–250 (Uniprot TNR16_Human) Transmembrane domain: aa 251–272 (Uniprot TNR16_Human) Cytosolic sequence: aa 273–275 (Uniprot TNR16_Human) ΔLNGFR was fused with P2A sequence (GGAAGCGGAGCTACTAACTTCAGCCTGCTGAAGCAGGCTGGCGACGTGGAGGAGAACCCTGGACCT) and followed by CAR sequence Signal peptide: aa 2–21 hCD8 (Uniprot CD8A_Human) Extracellular domain: anti-CD19 scFv (Clone: FMC 63); VL-G4S linker-VH Hinge/Transmembrane domain: aa 138–206 (Uniprot CD8A_Human) Co-stimulatory domain sequence: aa 180–220 (Uniprot CD28_Human) Signaling domain: aa 52–164 (Uniprot CD3Z_Human) |
| Recombinant DNA reagent | pLV-CD19-BBz CAR-G4S-mcherry | This paper | | Signal peptide: aa 1–21 (Uniprot CD8A_Human) Extracellular domain: anti-CD19 scFv (Clone: FMC 63) VL –G4S linker VH Hinge/Transmembrane domain: aa 138–206 (Uniprot CD8A_Human) Co-stimulatory domain sequence: aa 214–255 (Uniprot TNR9_Human) Signaling domain: aa 52–164 (Uniprot CD3Z_Human) CAR and mcherry was fused with G4S linker mCherry: aa 2–236 (Uniprot X5DSL3_ANAMA) |
| Recombinant DNA reagent | pLV-hCD19-G4S-Zsgreen | This paper | | Signal peptide: aa 1–19 (Uniprot CD19_Human) Extracellular domain: aa 20–291 (Uniprot CD19_Human) Cytoplasmic domain: aa314-556 (Uniprot CD19_Human) CD19 and Zsgreen was fused with G4S linker Zsgreen: aa 2–231 (Uniprot GFPL1_ZOASP) |
| Recombinant DNA reagent | pMDLg/pRRE | other | RRID:Addgene12251 | 3[rd] Lentivirus packaging vector |
| Recombinant DNA reagent | pRSV-Rev | other | RRID:Addgene12253 | 3[rd] Lentivirus packaging vector |
| Recombinant DNA reagent | pMD2.G | other | RRID:Addgene12259 | Lentivirus envelop vector |

*Continued on next page*

*Continued*

| Reagent type (species) or resource | Designation | Source or reference | Identifiers | Additional information |
|---|---|---|---|---|
| Recombinant DNA reagent | pMACS LNGFR | Miltenyi Biotec | Cat. #:130-091-890 | |
| Peptide, recombinant protein | rhIL-2 | BMI KOREA | | 300IU/mL |
| Peptide, recombinant protein | rhCD19-Fc | ACRO Biosystems | Cat. #: CD9-H5259 | FACS |
| Peptide, recombinant protein | AF647-conjugated streptavidin | Biolegend | Cat. #: 405237 | FACS |
| Antibody | InVivoMab anti-human CD3 | Bio X cell | Cat. #: BE0001-2-5MG | OKT3 clone 4 µg/mL coated |
| Antibody | InVivoMab anti-human CD28 | Bio X cell | Cat. #: BE0291-5MG | CD28.2 clone 2 µg/mL soluble |
| Antibody | hLNGFR-APC antibody | Miltenyi Biotec | Cat. #: 130-113-418 | FACS |
| Antibody | hCD19-APC antibody | Biolegend | Cat. #: 302212 | FACS |
| Other | DPBS | Welgene | Cat. #: LB001-02 | |
| Other | Fetal Bovine Serum (FBS) | Gibco | Cat. #: 26140–079 | |
| Other | DMEM | Gibco | Cat. #: 11965–118 | |
| Other | RPMI | Gibco | Cat. #: 21870–092 | |
| Other | HBSS with $Ca^{2+}$ and $Mg^{2+}$ | Gibco | Cat #: 14-025-092 | |
| Other | HEPES | Gibco | Cat. #: 1530080 | |
| Other | 2-Mercaptoethanol | Sigma | Cat. #: M6250-100mL | |
| Other | Lipofectamine 2000 | ThermoFisher | Cat. #: 11668019 | |
| Other | Glutamax | Gibco | Cat. #: 35050–061 | |
| Other | Protamine Sulfate | Sigma | Cat. #: P3369 | |
| Other | MEM Non-essential amino acid | Gibco | Cat. #: 111400500 | |
| Other | Sodium pyruvate | ThermoFisher | Cat. #: 11360070 | |
| Other | penicillin/streptomycin | Gibco | Cat. #: 15140–122 | |
| Other | Tomodish | Tomocube | | |
| Other | Paraformaldehyde Solution, 4% in PBS | ThermoFisher | Cat. #: AAJ19943K2 | Fixation |
| Other | CellBrite Fix 640 Membrane Dye | Biotium | Cat. #: 30089 | Plasma membrane staining |
| Other | VECTASHIELD Hardset w/ DAPI | VECTOR Laboratory | Cat. #: H-1400 | Mounting Medium |
| Other | LysoTracker Deep Red | ThermoFisher | Cat #: L12492 | Dye for labeling and tracking lytic granules |
| Other | TetraSpeck Microspheres,0.1 µm, fluorescent blue/green/orange/dark red | ThermoFisher | Cat #: T7279 | Fiducial markers |
| Other | FITC-dextran 10 kDa | TdB Labs | Cat #: 20682 | Fluorescein-labeled dextran, 10 kDa |
| Other | FITC-dextran 2000 kDa | TdB Labs | Cat #: 20584 | Fluorescein-labeled dextran, 2000 kDa |
| Commercial assay or kit | CD271 Microbeads kits, Human | Miltenyi Biotec | Cat. #: 130-099-023 | Isolation of CAR+ T cells |
| Commercial assay or kit | SepMate PBMC Isolation tube | STEMCELL | 86460 | PBMC Isolation |

*Continued on next page*

*Continued*

| Reagent type (species) or resource | Designation | Source or reference | Identifiers | Additional information |
|---|---|---|---|---|
| Software, algorithm | Flowjo | Flowjo | Version 10 | |
| Software, algorithm | Prism | GraphPad | Version 7 | |
| Software, algorithm | ImageJ | NIH | | |

## Primer list

| Primer name | Source or reference | Identifiers | Additional information (5'>3') |
|---|---|---|---|
| LNGFR F | This paper | PCR primers | GGGGATCCCCCCCATCAGTCCGCAAAG |
| LNGFR R-P2A | This paper | PCR primers | AGGTCCAGGGTTCTCCTCCACGTCGCCAGCCTGCTTC AGCAGGCTGAAGTTAGTAGCTCCGCTTCCCCACCTC TTGAAGGCTATGTAGG |
| P2A-CD19-BBz F | This paper | PCR primers | GGAAGCGGAGCTACTAACTTCAGCCTGCTGAAGCAG GCTGGCGACGTGGAGGAGAACCCTGGACCTGCCTTACCAG TGACCGCCTTGCTC |
| CD19 BBz R | This paper | PCR primers | TGTCGACTTAGCGAGGGGGCAGGGCCTGC |
| CD28/CD3 F | This paper | PCR primers | GTTATCACCCTTTACTGCAGGAGTAAGAGGAGCAGGCTC |
| CD28/CD3 R | This paper | PCR primers | GTCGACTTAGCGAGGGGGCAGGG |
| CD8Hinge/TM F | This paper | PCR primers | CAGTCACCGTCTCCTCAAC |
| CD8Hinge/TM R | This paper | PCR primers | GAGCCTGCTCCTCTTACTCCTGCAGTAAAGGGTGATAACCAG |
| mCherry F | This paper | PCR primers | GTGAGCAAGGGCGAGGAGGAT |
| mCherry-Sal1 R | This paper | PCR primers | TTGTCGACCTACTTGTACAGCTCGTCCATGCCGCCGG |
| G4S-mCherry F | This paper | PCR primers | GCTCCGGTGGTGGTGGTTCTGTGAGCAAGGGCGAGGAGGA TAAC |
| CD19 BBz CAR F | This paper | PCR primers | CCGGGGATCCATGGCCTTACCAGTGACCG |
| CD19 BBz CAR G4S R | This paper | PCR primers | AGAACCACCACCACCGGAGCCGCCGCCGCCAGAAC CACCACCACCGCGAGGGGGCAGGGCCTGCATGTGA |
| hCD19 F | This paper | PCR primers | GTTGGATCCATGCCACCTCCTCGCCTCCT |
| hCD19-G4S R | This paper | PCR primers | AGAACCACCACCACCGGAGCCGCCGCCGCCAG AACCACCACCACCCCTGGTGCTCCAGGTGCCCAT |
| G4S-Zsgreen F | This paper | PCR primers | GGTGGTGGTGGTTCTGGCGGCGGCGGCTCCG GTGGTGGTGGTTCTGCCACAACCATGGCCCAGTCCAAGC |
| Zsgreen R | This paper | PCR primers | GATTACGCGTATTGCTAGCTCAGGGCAAGGCGGAG |

## Cell lines and culture

K562 cells and CD19-positive K562 cells (K562-CD19; target cells) were kindly provided by Travis S. Young (California Institute for Biomedical Research). The original stock of K562 cells was purchased from American Type Culture Collection (ATCC). K562-CD19 cells were generated by transducing K562 cells (CD19-negative) with a lentivirus encoding human CD19. The cells were cultured in RPMI-1640 medium supplemented with 10% heat-inactivated fetal bovine serum (FBS), 2 mM L-glutamine, and 1% penicillin/streptomycin in a humidified incubator with a 5% $CO_2$ atmosphere at 37°C. The Lenti-X 293 T cell line was purchased from Takara Bio (Japan). The cells were maintained in

Dulbecco's modified Eagle medium supplemented with 10% heat-inactivated FBS, 2 mM L-gluta-mine, 0.1 mM non-essential amino acids, 1 mM sodium pyruvate, and 1% penicillin/streptomycin. We authenticated the identity of K562 and K562-CD19 cell lines using STR profiling, offered by Korean Cell Line Bank (KCIB). We also confirmed that K562 cell line are free from mycoplasma, which was performed by Korea Research Institute of Bioscience and Biotechnology (KRIBB). We attached the document for authentication (*Supplementary file 1*).

## Plasmid construction

CD19-specific chimeric antigen receptor (CD19-BBz CAR) was synthesized. The construct is composed of anti-CD19 scFv (FMC63) connected to a CD8α spacer domain and CD8α transmembrane domain, 4-1BB (CD137) co-stimulatory domains, and the CD3ζ signaling domain (*Rodgers et al., 2016*). The cytoplasmic domain comprised of a truncated CD271 (ΔLNGFR) gene for the isolation of T cells expressing CAR was amplified from pMACS-ΔLNGFR (Miltenyi Biotec, Germany) and over-lapped with the P2A oligonucleotide. The ΔLNGFR-P2A gene was overlapped with the CD19-BBz CAR gene and then inserted into the BamHI and SalI sites of pLV vectors to generate pLV- ΔLNGFR-P2A-CD19-BBz CAR. For the generation of CD19-CAR containing CD28 co-stimulatory domain (CD19-28z CAR), synthesized hCD28/CD3ζ fusion gene was overlapped with amplified CD8α spacer domain and CD8α transmembrane domain-containing PCR product. The final PCR product was then digested with SgrA1 and Sal1 and ligated into SgrA1 and Sal1-digested pLV- ΔLNGFR-P2A-CD19-BBz CAR vector.

To define the IS between CD19 expressing (K562-CD19) cells and CD19-specific CAR-T (CART19; effector) cells, we generated a mCherry-tagged CD19 BBz CAR and Zsgreen-tagged hCD19. The mCherry gene was amplified from pLV-EF1a-MCS-IRES-RFP-Puro (Biosettia, USA) and overlapped with synthetic oligonucleotides of a G4S linker. The CD19 BBz CAR gene was amplified with specific primer sets. The G4S-mcherry PCR product was overlapped with CD19 BBz CAR PCR product. The final PCR product was digested with BamH1 and Sal1 and then inserted into BamH1 and Sal1 digested pLV to generate pLV-CD19 BBz CAR-G4S mCherry. For the generation of Zsgreen-tagged hCD19 expressing vector, The Zsgreen gene was overlapped with synthetic oligonucleotides of a G4S linker. The G4S-Zsgreen PCR product was overlapped with a synthesized hCD19 (MN_001770.5) gene and inserted into the BamHI and MluI sites of pLV vectors to generate pLV-hCD19-G4S linker-Zsgreen.

## Generation of CAR-transduced human T cells

To generate a recombinant lentivirus supernatant, $6 \times 10^5$ Lenti-X 293 T cells were cultured in wells of a six-well plate for 24 hr and then transfected with the lentivirus packaging vectors (pMDL, pRev, pMDG.1) and the pLV vectors encoding ΔLNGFR-P2A-CD19-BBz CAR, ΔLNGFR-P2A-CD19-28z CAR, or mCherry-tagged CD19-BBz CAR using 10 μL of Lipofectamine2000 (Thermo Fisher Scientific, USA). Two days after transfection, the lentivirus containing supernatant was collected and stored at −80°C until used.

Peripheral blood mononuclear cells (PBMCs) were separated from whole blood samples of healthy donors using SepMate tubes (STEMCELL Technologies, Canada) following the manufacturer's instructions. The PBMCs were stimulated with 4 μg/mL of plate-bound anti-CD3 antibody (clone OKT3; Bio X cell), 2 μg/mL of soluble anti-CD28 antibody (clone CD28.2; Bio X cell), and 300 IU/mL human recombinant IL-2 (BMI KOREA, Republic of Korea).

Two days after stimulation, the activated T cells were mixed with the lentivirus supernatant, centrifuged at $1000 \times g$ for 90 min, and incubated overnight at 37°C. CAR-transduced T cells were cultured at a density of $1 \times 10^6$ cells/mL in RPMI-1640 supplemented with 10% heat-inactivated FBS, 2 mM L-glutamine, 0.1 mM non-essential amino acid, 1 mM sodium pyruvate, and 55 μM β–mercaptoethanol in the presence of human recombinant interleukin (IL)−2 (300 IU/mL) until isolation of CAR-expressing T cells from bulk T cells.

The percentage of CAR and ΔLNGFR-positive T cells was assessed by biotin-conjugated rhCD19-Fc (Cat # CD9-H5259, ACRO Biosystems, USA) with AF647-conjugated streptavidin (Cat # 405237, Biolegend, USA), and fluorescein isothiocyanate (FITC)-conjugated LNGFR antibody (Cat# 130-112-605, Miltenyi Biotec, Germany).

## Isolation of CAR-transduced T cells

CAR- and ΔLNGFR-positive T cells were isolated using the human CD271 MicroBead kit (Cat# 130-099-023, Miltenyi Biotec) following the manufacturer's instructions. Sorted CART19 cells were expanded for six days with RPMI-1640 medium supplemented with 10% heat-inactivated FBS, 2 mM L-glutamine, 0.1 mM non-essential amino acids, 1 mM sodium pyruvate, and 55 µM β–mercaptoethanol in the presence of recombinant human rhIL-2 (300 IU/mL).

## Lentiviral engineering of Zsgreen-tagged hCD19 expressing cell lines

K562-CD19 cell lines stably expressing Zsgreen were generated by lentiviral transduction with supernatant containing the Zsgreen. hCD19-G4S-linker-Zsgreen overexpressing K562 (K562-CD19-G4S-Zsgreen) cell lines were produced by lentiviral transduction with supernatant containing the hCD19-G4S-linker-Zsgreen. Two days after transduction, transduced cells were stained using hCD19 specific antibody (Biolegend, Clone HIB19) and analyzed with flow cytometry. We confirmed that the percentage of CD19$^+$Zsgreen$^+$ cells was nearly 98%. (*Figure 8—figure supplement 1*).

## Sample preparation for imaging

The effector or target cells were diluted at 600 cells/µL, respectively. We used a petri dish compatible with our experimental setup (TomoDish, Tomocube Inc, Republic of Korea). 12 µL of each diluted cell was seeded on TomoDish, gently mixed, and covered with a square cover-slip glass (22 × 22 mm$^2$, No. 1.5H, Deckglaser Inc). The side of the dish was sealed by 10 µL of mineral oil (M-8410, Sigma), which prevented introduction of exterior contaminants and drying of the medium.

## Membrane staining

To image cell membranes using fluorescence microscopy, effector and target cells were respectively stained with 1X CellBrite Fix 640 Membrane Dye (Biotium) for 15 min at 37℃, twice washed with DPBS, and then suspended using RPMI medium. 2 × 10$^5$ cells of each effector and target were seeded on a Tomodish and incubated for 15 min at 37℃. Then the medium was replaced with 4% paraformaldehyde (PFA) solution to fix them. After 10 min, 4% PFA was removed and twice washed with DPBS. For membrane staining, the cells were mounted with 25 µL VECTASHIELD Antifade mounting medium with DAPI (VECTOR laboratory, H-1200).

## Dextran solution

To image the synaptic cleft, we used FITC-labeled dextran of molecular weights 10 and 2000 kDa (TdB Labs), whose reported hydrodynamic diameters were 4 and 54 nm, respectively. We diluted the dextran in PBS and the final concentration was 50 µM for the 4 nm dextran and 2.5 µM for the 54 nm dextran.

## Lysotracker staining

To image the lytic granules of CART19 cells using fluorescence microscopy, CART19 cells were stained with 75 nM Lysotracker Deep Red (Thermofisher) solution in 1X Hanks' Balanced Salt solution (HBSS; Thermofisher) containing Ca$^{2+}$ and Mg$^{2+}$ ions for 50 min at 37℃, washed with HBSS, and then suspended using complete T cell medium. Similar to the imaging protocol for membrane-stained cells, the stained CART19 cells were mixed with K562-CD19 cells on a Tomodish. To image the transport dynamics of the lytic granules using widefield deconvolution microscopy, the mixed cells were imaged in live states. To image the CAR IS using 3D-SIM, the mixed cells were incubated for 15 min at 37℃, and chemically fixed by replacing the medium with 4% paraformaldehyde (PFA). After 10 min, 4% PFA was removed, washed and replaced with complete T cell medium.

## Correlative fluorescence microscopy

To evaluate the segmentation performance quantitatively, the evaluation data were prepared using a custom-built setup for correlative imaging between wide-field fluorescence microscopy and ODT (*Kim et al., 2017*). CART19 (mCherry) and K562-CD19 (Zsgreen) cells, and their plasma membranes were imaged in different fluorescence channels, respectively. To excite mCherry and GFP proteins and the plasma membrane of the prepared cells, blue (wavelength = 488 nm, Cobolt, 06-MLD), green (wavelength = 561 nm, CNI Laser, MLL-FN-561), and red (wavelength = 639 nm, CNI Laser,

MLL-FN-639) DPSS lasers were illuminated in wide-field epi-fluorescence geometry. 3D fluorescence image stacks were obtained by scanning the objective lens with an axial spacing of 300 nm and imaged with an sCMOS camera (Neo 5.5 sCMOS, Andor Technology). The obtained fluorescence images were deconvolved using the blind Lucy algorithm (a *deconvblind* function in MATLAB) for ten maximal iterations with theoretical 3D point spread functions as initial estimates. The deconvoled fluorescence images were registered with RI tomograms, and the ground-truth labels of the CART19 and K562-CD19 cells were derived from expert biologists who manually thresholded, delineated, and smoothed the cell volume by means of the overlapped RI and fluorescence images.

### 3D confocal fluorescence microscopy

To resolve the details of the membrane topology of the IS between mCherry-tagged CD19-CAR-expressing T cells and Zsgreen-tagged CD19 expressing K562 cells, 3D confocal fluorescence microscopy was performed. The membrane topology of IS was analyzed with a commercial confocal microscope (Nikon Eclipse Ti) and a high-NA objective lens (Nikon Apo 60×, 1.4 NA) to obtain a high-resolution, multicolor, 3D fluorescence image.

### Complementary 3D-SIM and ODT

For complementary imaging of ODT with high-resolution fluorescence microscopy, 3D-SIM was integrated in a custom-built setup based on a digital micromirror device (*Figure 6—figure supplement 1*; *Shin et al., 2018*). A polarizing beam splitter separated scanning plane waves for ODT and structured illumination patterns for 3D-SIM into the transmitted and epi-illumination mode, respectively. A dichroic mirror combined the ODT and 3D-SIM signals, and a 10:90 beam splitter divided the merged signals to an ODT camera (MQ042MG-CM, Ximea) and a 3D-SIM camera (Panda 4.2, PCO) respectively. The raw 3D-SIM image stacks were acquired with five pattern phases spaced by $2\pi/5$, three pattern orientations spaced 60° apart, and axial translation of an objective lens equipped with a piezoelectric Z stage (P-721.CDQ, Physik Instrumente), at the interval of 45–50 nm for 100-nm-diameter Tetraspeck beads (ThermoFisher) and 180 nm for CART19/K562-CD19 conjugates respectively. The 3D-SIM images were then reconstructed using the custom MATLAB codes based on Wiener deconvolution and Richardson-Lucy deconvolution for the beads and the cells, respectively (*Brown et al., 2011*; *Müller et al., 2016*).

### Design of DCNN architecture

The DCNN architecture was designed on the basis of UNet architecture, which features excellent performance for various biomedical volumetric segmentation tasks such as multi-cell (*Ronneberger et al., 2015*), organ (*Roth, 2017*), and tumor segmentation tasks (*Dong et al., 2017*). Our model employed five contracting and expanding layers comprising 32, 64, 128, 256, and 512 filters, respectively. To improve the segmentation performance, several modifications of the architecture were added while maintaining the overall U-shaped feature map flow line. First, we employed a series of ResNet blocks from ResNet (*He et al., 2016*) in the contracting paths for extracting the feature more robustly. Also, to increase the receptive field, we employed the feature skip connection that passes through the global convolutional network layer (*Peng et al., 2017*) with $k$ = 13, 13, 9, 7, and 5, respectively. The overall schematic figure of DCNN architecture is shown in *Figure 2—figure supplement 3*. Our network was implemented in Python using the PyTorch package (http://pytorch.org), and the processing steps were performed in MATLAB (MathWorks, Inc).

### Statistical analysis

MATLAB was used in order to compare the sample means by two-tail paired Wilcoxon tests in *Figure 8g* and two-tail unpaired Wilcoxon tests in *Figure 9*, *Figure 2—figure supplement 2b*, and *Figure 6—figure supplement 2c* respectively. All of the numbers following the ± sign in the text are standard deviations.

### Major datasets and codes

We have provided pre-processing and post-processing codes, and training and untrained datasets used in *Figure 3—video 1* (https://osf.io/9w32p/). Also, the DeepIS code and the processing codes

are available through a GNU General Public License at https://github.com/JinyeopSong/2020__
DeepIS (*Song and Lee, 2020*).

## Acknowledgements

This work was supported by KAIST Up program, Tomocube Inc, National Research Foundation of Korea (NRF) (2017M3C1A3013923, 2015R1A3A2066550, 2018K000396, NRF-2019R1A2C1004129), the Bio and Medical Technology Development Program of NRF funded by the Ministry of Science & ICT (2014M3A9D8032525), KAIST GCORE(Global Center for Open Research with Enterprise) grant funded by the Ministry of Science and ICT (N11190028).

## Additional information

### Competing interests

Moosung Lee: Mr. Moosung Lee has financial interests in Tomocube Inc, a company that commercializes optical diffraction tomography and quantitative phase-imaging instruments, and is one of the sponsors of the work. Young-Ho Lee: Dr. Y.H. Lee is an employee of Curocell Inc. Chan Hyuk Kim: Prof. C. H. K. is a co-founder and shareholder of Curocell inc. YongKeun Park: Prof. Park has financial interests in Tomocube Inc, a company that commercializes optical diffraction tomography and quantitative phase-imaging instruments, and is one of the sponsors of the work. The other authors declare that no competing interests exist.

### Funding

| Funder | Grant reference number | Author |
|---|---|---|
| National Research Foundation of Korea | 2017M3C1A3013923 | Moosung Lee<br>Jinyeop Song<br>Geon Kim<br>YongKeun Park |
| National Research Foundation of Korea | 2015R1A3A2066550 | Moosung Lee<br>Jinyeop Song<br>Geon Kim<br>YongKeun Park |
| National Research Foundation of Korea | 2018K000396 | Moosung Lee<br>Jinyeop Song<br>Geon Kim<br>YongKeun Park |
| The Ministry of Science and ICT | 2014M3A9D8032525 | Young-Ho Lee<br>Chan Hyuk Kim |
| The Ministry of Science and ICT | N11190028 | Young-Ho Lee<br>Chan Hyuk Kim |
| National Research Foundation of Korea | 2019R1A2C1004129 | Young-Ho Lee<br>Chan Hyuk Kim |
| KAIST | Up prorgam | YongKeun Park |

The funders had no role in study design, data collection and interpretation, or the decision to submit the work for publication.

### Author contributions

Moosung Lee, Conceptualization, Data curation, Software, Formal analysis, Validation, Investigation, Visualization, Methodology, Project administration; Young-Ho Lee, Conceptualization, Resources, Software, Supervision, Funding acquisition, Validation, Investigation, Project administration; Jinyeop Song, Conceptualization, Resources, Data curation, Software, Formal analysis, Validation, Investigation, Methodology, Project administration; Geon Kim, Conceptualization, Resources, Data curation, Software, Formal analysis, Supervision, Funding acquisition, Validation, Investigation, Methodology, Project administration; YoungJu Jo, YongKeun Park, Conceptualization, Resources, Software,

Supervision, Funding acquisition, Validation, Project administration; HyunSeok Min, Supervision, Validation, Methodology; Chan Hyuk Kim, Conceptualization, Supervision, Funding acquisition, Validation, Project administration

### Author ORCIDs
Moosung Lee (ID) https://orcid.org/0000-0002-2826-5401
YongKeun Park (ID) https://orcid.org/0000-0003-0528-6661

### Decision letter and Author response
Decision letter https://doi.org/10.7554/eLife.49023.sa1
Author response https://doi.org/10.7554/eLife.49023.sa2

## Additional files
### Supplementary files
• Supplementary file 1. Documents for cell line authentication.

• Transparent reporting form

### Data availability
We have provided pre-processing and post-processing codes, and training and validation datasets used in Figure 3-Video 1 (https://osf.io/9w32p/). Also, the Unet architecture code is available in https://github.com/JinyeopSong/190124_CART-Segmentation-best (copy archived at https://archive.softwareheritage.org/swh:1:rev:eca787f7cc0b3aa423c54ce3ac53088e6049948b/).

The following dataset was generated:

| Author(s) | Year | Dataset title | Dataset URL | Database and Identifier |
|---|---|---|---|---|
| Lee M | 2019 | DeepIS: deep learning framework for three-dimensional label-free tracking of immunological synapses | https://osf.io/9w32p/ | Open Science Framework, 9w32p |

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
