## [Decision Letter]

**Acceptance summary:**

Your manuscript demonstrates the power of using label free 3D imaging, provided by optical diffraction tomography, for long term studying of intercellular dynamical process. The example of imaging and analyzing the immunological synapse (IS) is potentially impactful, as it offers new avenues in studying immune responses and their downstream signalling events. The addition of deep convolutional neural network for 3D segmentation and resolving cell to cell interactions is also important to enable real time observation of synaptic events. This work may stimulate more interesting and unexplored biomedical applications using optical diffraction tomography. You have also identified approaches to more fully capture the power of the deep learning by removing subjective elements of the training process.

**Decision letter after peer review:**

Thank you for submitting your article "DeepIS: deep learning framework for three-dimensional label-free tracking of immunological synapses" for consideration by *eLife*. Your article has been reviewed by three peer reviewers, including Michael L Dustin as the Reviewing Editor and Reviewer #1, and the evaluation has been overseen by Satyajit Rath as the Senior Editor. The following individuals involved in review of your submission have agreed to reveal their identity: Christoph Wülfing (Reviewer #2); Klaus Ley (Reviewer #3).

The reviewers have discussed the reviews with one another and the Reviewing Editor has drafted this decision to help you prepare a revised submission. *eLife*

There is agreement among reviewers that the use of deep learning to segment cell pairs based on holographic Optical diffraction tomography (Tomocube) is potentially useful for analysis of immunological synapses and other situations where cell-cell interfaces with particular refractive index contrast are generated. The method needs to be better validated with an objective segmentation approach and a few other issues regarding utility of the data better elaborated.

Essential revisions

1) It remains somewhat unclear how (and whether) membrane topology at the cellular interface is determined by Lee et al. and this should be explained in more detail. Looking at representative ODT data, e.g. in Figure 1C and Figure 2A, the cellular interface seems sufficiently tight that the T cell and the APC can't be distinguished unambiguously in the ODT images. This is consistent with electron microscopy data showing membrane apposition below the resolution limit of light microscopy. Therefore, the interface topology had to be inferred. This was done by training neural networks with manually curated data on 236 T cell/APC couples. The authors should describe how the interface was manually identified. Which quantitative criteria were used? The authors should show a number of manually annotated cell couples as raw data and with the manually annotated interface. T cell/APC interfaces can be highly undulating. Were such undulations resolved? Based on the representative ODT data that seems unlikely. From Figure 2B it looks like the interface was estimated as the surface that minimises distances from the centres of each cell. Because of membrane undulations, this would likely be biologically incorrect. Without reliable determination of the cellular interface ODT data on cell couples will remain highly limited.

2) Two-colour fluorescence data are used to validate the segmentation of T cell and APC in Figure 5. T cells are identified with a CAR-mCherry fusion protein, the APCs with actin-GFP. The fusion proteins are well suited to identify the cell types. However, as the CAR-mCherry fusion protein seems predominantly vesicular, its fluorescence is ill-suited to resolve membrane topology as one of the most important potential types of data derived by ODT. A non-transferring plasma membrane localised protein or lipid should be used instead to demarcate the membrane of each cell. If the authors can then access 3D STED or a localization method like double helix this may be their best bet to be able to get enough information about membrane topology to trace the interface in an inambiguous manner and validate the limits of the method. The authors may also want to use a lysotracker dye to identify the dense core granules and an extracellular dye with confocal microscopy sectioning to define the synaptic cleft to determine the conditions under which the method can identify these features.

3) An alternative approach to validation would be correlative electron microscopy. The most readily correlated data might be obtained by an ion beam milling system as the context of the cell on a indexed surface could be recorded in the ODT imaging and then use to find the same cell pair and map the interface in 3D. If this was done on a few cells, the training data could be potentially enable the machine learning algorithm to exceed a human in assessing how information in the ODT data set relates to the actual interface. While revisions in *eLife* are typically restricted to 2 months, if arranging access to such a system required more time and Lee et al. want to undertake it, we could provide an extension.

4) What do the data in Figure 6 tell the reader about how CAR T cells function? As shown, the data simply state that CAR T cells need engagement of the CAR to form a cell couple, something a much more basic cell coupling assay can establish. Is there any biological information in the morphological and refractive index changes observed? The authors may want to consider using different T cell activation conditions that both lead to cell coupling, e.g. comparing two CARs with different signalling motifs, to see whether they can, at the least, detect significant differences in morphological and refractive index parameters or better relate such changes to cellular function.

[Editors' note: further revisions were suggested prior to acceptance, as described below.]

Thank you for submitting your article "Deep-learning based three-dimensional label-free tracking and analysis of immunological synapses of CAR-T cells" for consideration by *eLife*.

The Senior and Reviewing Editor have carefully evaluated your resubmission and while a number of issues are addressed, an essential issue related to using a higher resolution methods to validate the interface area and composition and/or improve the training based on this information was not addressed. Recent studies suggest that membrane infolding and release of granules into the interface are important events and it would be critical to determine if there is enough information in the tomograms to train the system to correctly identify these events. The use of the system as it stands will not provide sufficiently robust output to be of broad utility and will be better published in a more specialised journal. If this essential revision can be addressed you can resubmit using the link below, but if not you should submit the generally improved paper to another journal.

[Editors' note: further revisions were suggested prior to acceptance, as described below.]

Thank you for submitting your article "Deep-learning based three-dimensional label-free tracking and analysis of immunological synapses of CAR-T cells" for consideration by *eLife*. Your article has been reviewed by three peer reviewers, including Michael L Dustin as the Reviewing Editor and Reviewer #1, and the evaluation has been overseen by Satyajit Rath as the Senior Editor. The following individuals involved in review of your submission have agreed to reveal their identity: Christoph Wülfing (Reviewer #2); Renjie Zhou (Reviewer #4).

The reviewers have discussed the reviews with one another and the Reviewing Editor has drafted this decision to help you prepare a revised submission.

Summary:

The reviewers all appreciate the work put into the revision. It was suggested that correlative electron microscopy was needed to establish ground truth for training and assessment of accuracy, but instead you provide 3D SIM data on various cellular molecules that unfortunately have their own complex localisations and may not really reflect the actual synaptic cleft between the cells. The reviewers can make one further suggestion requiring some experiments the could be used to test the accuracy of the trained DeepIS algorithm at 3D SIM resolution limit of ~125 nm.

Essential revisions:

1) In 2014 Dan Davis' lab developed a negative stain method using fluorescent dyes in the media to define the location of the immunological synapse using confocal microscopy. See: Cartwright AN, Griggs J, Davis DM. The immune synapse clears and excludes molecules above a size threshold. Nat Commun. 2014;5:5479. Epub 2014/11/20. doi: 10.1038/ncomms6479. PubMed PMID: 25407222; PMCID: PMC4248232. It should be possible to use this method with 3D SIM or perhaps Airyscan confocal microscopy to define the location of "synaptic cleft" between the Car-T and target with 125 nm resolution. Correlative ODT scored with Deep IS to define the synapse could then be quantitatively compared to the 3D SIM/confocal analysis of the synaptic cleft to objectively assess the accuracy of deep IS to a ground truth measurement of greater resolution. See below discussion for resolution that would make the 3D SIM a reasonable ground truth for the ODT data sets. The reviewers would accept this approach as an alternative to electron microscopy.

2) For the resolution definition in ODT, you appear to be using the Nyquist sampling period as the imaging resolution. To resolve structures, we need the sample to be separated by at least 2 Nyquist sampling periods. Therefore, the optical resolution should be 2x of the Nyquist resolution you have used in the manuscript. This definition is also consistent with the Abbe resolution definition. Therefore, the lateral resolution in ODT should be λ/2NA which is more than 200 nm not 125 nm. Similarly, the axial resolution should be corrected. If you insist on using Nyquist sampling period as resolution (which seems incorrect), you should clarify it and compare it with the conventionally used Abbe resolution limit. You should also see if you can obtain a 3D standard sample, like an Argolight slide, to directly determine the resolution of the ODT system in a more direct manner.

3) Aside from resolution, there is also a concerned about the refractive index (RI) sensitivity of ODT. When structures have very small (RI) contrast, it becomes a sensitivity issue more than a resolution issue, e.g., ODT is not able to detect many organelles or granules in the cells even when their diameters are larger than 200 nm (the resolution limit). Could the authors provide an estimation of RI contrast between the membranes and estimate whether they have the detection sensitivity for detecting granule release events. With regard to RI, you continue to interpret the RI in terms of proteins density and this this seems overly simplistic as area within the resolution limit of the interface will contain many types of structures. So this interpretation should be toned down to more of a hypothetical argument about average macromolecule density given convolved RI of lipids, proteins and carbohydrates in the "synapse".

[Editors' note: further revisions were suggested prior to acceptance, as described below.]

Thank you for resubmitting your work entitled "Deep-learning based three-dimensional label-free tracking and analysis of immunological synapses of CAR-T cells" for further consideration by *eLife*. Your revised article has been evaluated by Satyajit Rath (Senior Editor) and a Reviewing Editor.

The manuscript demonstrates the power of using label-free 3D imaging, provided by optical diffraction tomography, for long-term studying of intercellular dynamical process. The example of imaging and analyzing the immunological synapse (IS) is potentially impactful, as it offers new avenues in studying immune responses and their downstream signaling events. The addition of deep convolutional neural network for 3D segmentation and resolving cell to cell interactions is also important to enable real-time observation of synaptic events. This work may stimulate more interesting and unexplored biomedical applications using optical diffraction tomography. There are 3 remaining issues that need to be addressed before acceptance, as outlined below:

1) The Results still state that “lymphocytes contain lipid-rich environment localized mostly on a 4-nm-thick membrane site, whose size is beyond optical resolution and implies a negligibly small amount of lipid molecules compared with proteins.” The reviewers and editor agree that the membrane lipid and protein contributions to the refractive index in relation to ODT is likely to be small for the reason you state, with the major contribution to the pixel intensity being cytoplasmic protein and cortical cytoskeleton. Can you rewrite this passage to clarify the situation that the issue is not so much lipid vs protein, but sub resolved membrane vs the relatively voluminous adjacent cytoplasm.

2) The error of synaptic cleft determination is on the order of 0.5 µm and this is a serious limitation currently as it suggest the ability of an expert trainer to correctly identify the synaptic cleft from ODT images is poor. Can you discuss the idea training using the negative staining with the smallest dextran as a ground truth might further improve the power of the approach by eliminating the training error. Then if the ODT contains subtle information that the algorithm can can be trained to exploit, this information will be delivered with not bias other than the resolution limit. Then you would need much more correlative negative stain-ODT for training set and a distinct testing set to determine if there is an improvement. We are not asking you to do this, just to discuss it as a possible way to improve the method in the future.

3) The statement in the Abstract that you can't do long term 3D fluorescence microscopy has been made false in recent years by lattice light sheet microscopes. While these are not widely available, they will be in future years based on the new Zeiss LLSM7 release. Please rewrite the Abstract to acknowledge lattice light sheet microscopy as a competing technology and perhaps discuss how ODT with analysis trained to identify the synapse will offer complementary information to LLSM.

---

## [Author Response]

Essential revisions1) It remains somewhat unclear how (and whether) membrane topology at the cellular interface is determined by Lee et al. and this should be explained in more detail. Looking at representative ODT data, e.g. in Figure 1C and Figure 2A, the cellular interface seems sufficiently tight that the T cell and the APC can't be distinguished unambiguously in the ODT images. This is consistent with electron microscopy data showing membrane apposition below the resolution limit of light microscopy. Therefore, the interface topology had to be inferred. This was done by training neural networks with manually curated data on 236 T cell/APC couples. The authors should describe how the interface was manually identified. Which quantitative criteria were used? The authors should show a number of manually annotated cell couples as raw data and with the manually annotated interface.

We thank the reviewers for the valuable comment. To address this issue, the performance of our segmentation framework is supplemented in the revised manuscript. In addition, relevant datasets and processing codes are provided (https://osf.io/9w32p/).

Here we further explain the details on our annotation method with manual parameter selections (Figure 2—figure supplement 1). The watershed segmentation on a distance-transformed mask caused an incorrect boundary definition (Figure 2—figure supplement 1A). To correct the ill boundary definition, we applied the watershed segmentation to the Gaussian-smoothed RI tomogram after iterative parameter adjustment (Figure 2—figure supplement 1B). Because the RI tomogram distinguished cell boundaries, adjoining cells could be segmented successfully. Although this process produced over-segmented fragments due to the presence of multiple local maxima, we could combine them into effector and target masks after comparing them with the original RI tomogram. A number of annotated masks are shown in Figure 3—video 1.

Next, we verified that human raters (three experts in cell biology) who participated in the data curation performed consistent and good segmentation performance. This was achieved by measuring the segmentation sensitivity using simultaneous truth and performance level estimation (STAPLE), which quantified the inter-rater variability of annotation performance (10.1109/TMI.2004.828354). Specifically, we asked the experts to manually draw each cell mask from representative cross-section images of RI maps (10 from curated dataset and 10 from untrained dataset), and used STAPLE algorithm to estimate the truth masks and inter-rater sensitivity.

For training and untrained datasets, we compared the representative STAPLE truth estimates with the annotation masks (Figure 2—figure supplement 2A). As expected, the two masks were sufficiently similar for both the training and untrained datasets. The similarity between the annotation masks and STAPLE truth estimates was quantitatively compared by statistical analysis of Pearson correlation coefficients ( Figure 2—figure supplement 2B). The comparison showed higher mean and smaller standard deviation values of the coefficient in the training dataset, indicating that the quality of curated annotation masks reached the expert levels. For both training and untrained datasets, we also validated the consistent annotation performance of the experts from their high inter-rater sensitivity (> 97.5%) (Figure 2—figure supplement 2C). Based on the performance validation, the experts heuristically curated well-segmented data with consensus.

T cell/APC interfaces can be highly undulating. Were such undulations resolved? Based on the representative ODT data that seems unlikely. From Figure 2B it looks like the interface was estimated as the surface that minimises distances from the centres of each cell. Because of membrane undulations, this would likely be biologically incorrect. Without reliable determination of the cellular interface ODT data on cell couples will remain highly limited.

We show that DeepIS resolved undulation problems by displaying 22 individual temporal synapse area curves used in Figure 6B (Figure 8—figure supplement 1). The smooth change in overall data suggests that our method resolved undulation problems in most cases. Several spikes attest to the fact that the method suffers from ill segmentations in some cases. Although these outlier problems could be alleviated by post-processing such as data blocking, we used the raw data for analysis because the number of ill-segmented data points was extremely small (5 out of 1329 data points for 200 seconds; 0.38%). Our supplementary videos also corroborate these results. Overall, our approach provided sufficient fidelity for the quantitative analysis of IS dynamics.

2) Two-colour fluorescence data are used to validate the segmentation of T cell and APC in Figure 5. T cells are identified with a CAR-mCherry fusion protein, the APCs with actin-GFP. The fusion proteins are well suited to identify the cell types. However, as the CAR-mCherry fusion protein seems predominantly vesicular, its fluorescence is ill-suited to resolve membrane topology as one of the most important potential types of data derived by ODT. A non-transferring plasma membrane localised protein or lipid should be used instead to demarcate the membrane of each cell. If the authors can then access 3D STED or a localization method like double helix this may be their best bet to be able to get enough information about membrane topology to trace the interface in an inambiguous manner and validate the limits of the method. The authors may also want to use a lysotracker dye to identify the dense core granules and an extracellular dye with confocal microscopy sectioning to define the synaptic cleft to determine the conditions under which the method can identify these features.

We thank the reviewers for the valuable suggestion. The raised question concerns whether CAR-mCherry is suitable for resolving the membrane topology. This issue might arise due to the fluorescence images in Figure 5, where the CAR fluorescence intensity on the plasma membrane was not clearly visible. Thus, we performed extended experiments to supplement the results in Figure 5. The overall results validated that RI image and CAR-mCherry fusion protein can adequately resolve the membrane topology of CART19 cells and our manual labeling method in Figure 5 was valid.

First, we elucidated the distribution of CAR proteins with a higher spatial resolution and sensitivity than epi-fluorescence microscopy. Among many suggestions by the reviewers, we used a multicolor 3D confocal microscope (Nikon Eclipse Ti, Nikon Apo 60×, 1.4 NA) because it already provided sufficient 3D spatial resolution to image overall structures. The imaged cells were fixed conjugate pairs of CART19-CAR-G4S-mCherry and K562-CD19-G4S-Zsgreen cells. To visualize the plasma membranes and nuclei of the cells, we stained the cells with APC-conjugated plasma membrane staining dye and DAPI, respectively. The resultant fluorescence image showed that the cells were successfully stained (Figure 5—figure supplement 2A). In particular, CART19 and K562-CD19 cells expressed CAR and CD19 respectively mostly on the plasma membrane. This was also true when a CART19 cell was conjugate to a K562-CD19 cell (Figure 5—figure supplement 2B-D). The 3D view confirmed that CAR was coalesced around the immunological synapse and also spread throughout the plasma membrane. Therefore, fluorescence imaging of CAR was sufficient to resolve the membrane topology although it was vesicular.

We then performed an additional experiment to demonstrate that RI data provides essential information about the cellular topology. For 3D correlative RI and fluorescence imaging, we used a custom-built 3D correlative fluorescence microscopy setup which was equipped with an sCMOS camera (Neo 5.5 sCMOS, Andor Technology). As suggested by the reviewers, we performed correlative analysis of membraned-stained CART19/K562-CD19 conjugate pairs (Figure 5—figure supplement 1). The membrane fluorescence images overlapped well with the RI borders and the IS interfaces co-localized by CAR and CD19 proteins (Figure 5—figure supplement 1A). Based on the correlative images, CART19 and K562-CD19 cell labels were manually delineated and compared with the DeepIS results (Figure 5—figure supplement 1B). Both Pearson correlation coefficients and boundary displacement errors were improved with the enhanced image quality.

The confocal and 3D correlative images convinced us that RI and CAR-mCherry signals can sufficiently resolve the membrane topology without membrane staining under improved imaging conditions. To verify this, we finally performed additional correlative analysis without membrane staining (Figure 5A). The resulting images provided clear distinctions between CART19 and K562 cells. Also, the remaining membrane structures of CART19 cells could be manually delineated using the correlative images. The accuracy parameters were also comparable to those of membrane-stained cells (Figure 5B). In conclusion, our previous manual labeling procedure was sufficient to demarcate the membrane structures of CART19 cells. These contents have been added to the revised manuscript.

3) An alternative approach to validation would be correlative electron microscopy. The most readily correlated data might be obtained by an ion beam milling system as the context of the cell on a indexed surface could be recorded in the ODT imaging and then use to find the same cell pair and map the interface in 3D. If this was done on a few cells, the training data could be potentially enable the machine learning algorithm to exceed a human in assessing how information in the ODT data set relates to the actual interface. While revisions in eLife are typically restricted to 2 months, if arranging access to such a system required more time and Lee et al. want to undertake it, we could provide an extension.

Although the suggested correlative electron microscopy is a scintillating approach, we consider that these experiments are far beyond the scope of current work. Instead, we focus on other supplementary experiments to address the remaining relevant issues. We believe that this choice will make the manuscript deliver a coherent message and better quality. The suggested idea will be a very interesting topic for the following studies.

4) What do the data in Figure 6 tell the reader about how CAR T cells function? As shown, the data simply state that CAR T cells need engagement of the CAR to form a cell couple, something a much more basic cell coupling assay can establish. Is there any biological information in the morphological and refractive index changes observed? The authors may want to consider using different T cell activation conditions that both lead to cell coupling, e.g. comparing two CARs with different signalling motifs, to see whether they can, at the least, detect significant differences in morphological and refractive index parameters or better relate such changes to cellular function.

We sincerely appreciate the critical comment. The most important aspect of Figure 6 is that we have established early IS kinetics by CART19 cells quantitatively without fluorescence microscopy. Using 3D high-speed ODT and deep learning, we concluded that both IS areas and surface protein density of CAR T cells reached a steady-state within a few minutes. As described in the original manuscript, these observations connect previous studies based on various fluorescence microscopy techniques, which revealed the protein aggregation at the IS and mechanochemical potentiation and centrosome polarization of T cells during early IS dynamics.

We nonetheless agree that the question raised by the reviewers is also relevant to the future biological applications of our approach. Particularly interesting is to understand whether the types of co-stimulatory domains affect CAR IS characteristics, because these studies may provide insights into the efficacy of different therapeutic CAR agents. For example, Kymriah and Yescarta are the only two CD19-targeting CAR-T cell therapies that have been approved by US-FDA to date. Since one of the major differences between the two 2^nd^ generation CAR-T cells lies in the sequence of the costimulatory signaling domain used (4-1BB for Kymriah and CD28 for Yescarta), we thought to apply our method to compare the IS characteristics of the CAR-T cells with different costimulatory signaling domains. Primary human T cells transduced with the lentiviral vectors encoding CD19-28z or CD19-BBz CAR showed comparable transduction efficiency as well as similar surface expression levels of CARs as determined by flow cytometry (Figure 7—figure supplement 1a).

Exploiting our established method, we studied the effect of CD28 or 4-1BB co-stimulatory domains on CAR IS characteristics. We first quantified the differences of early IS kinetics between CD28 and 4-1BB based CAR-T cells in the presence of K562-CD19 target cells and found stable IS formation within several minutes, regardless of the types of costimulatory domains (Figure 7—video 1). Quantitative analysis also indicated no statistically significant differences in early IS kinetics between CD19-28z and CD19-BBz CAR-T cells (Figure 9—figure supplement 2). We then compared the statistics of other IS parameters between the CD19-28z and CD19-BBz CAR-T cells in a steady state. Specifically, we incubated each type of CAR-T cells with K562-CD19 cells for 15 minutes to allow sufficient time for stable IS formation, and fixed them with 4% paraformaldehyde solution. When we analyzed the images for conjugates (Figure 9A), we found no significant difference in IS areas between the two CAR-T cell types (Figure R8b). However, interestingly, significantly higher IS protein densities and total IS protein amounts were observed for CD19-BBz CAR-T cells, by approximately 10% as determined by quantitative statistical analyses, compared with CD19-28z CAR-T cells (Figures 9C-D).

Collectively, our results indicate that the quantitative analysis of IS parameters using DeepIS, in conjunction with other analytical methods such as fluorescence-based microscopy and quantitative mass-spectrometry, may help to elucidate the mechanistic details underlying the functional differences observed for the CAR-T cells with different signaling domains. Because this finding is significant, we have added this result as the main result (Figure 7) in the revised manuscript.

[Editors' note: further revisions were suggested prior to acceptance, as described below.]

The Senior and Reviewing Editor have carefully evaluated your resubmission and while a number of issues are addressed, an essential issue related to using a higher resolution methods to validate the interface area and composition and/or improve the training based on this information was not addressed. Recent studies suggest that membrane infolding and release of granules into the interface are important events and it would be critical to determine if there is enough information in the tomograms to train the system to correctly identify these events. The use of the system as it stands will not provide sufficiently robust output to be of broad utility and will be better published in a more specialised journal. If this essential revision can be addressed you can resubmit using the link below, but if not you should submit the generally improved paper to another journal.

We thank all the reviewer for the helpful suggestions. The important remaining issue is to compare the immunological synapse (IS) obtained by our method with a conventional high-resolution method for conclusive validation. To address the reviewer’s comment, we have integrated three-dimensional structured illumination microscopy (3D-SIM) for high-resolution fluorescence microscopy to the used ODT setup (Figure 6—figure supplement 1A). The resolution improvement was validated by reconstructing the SIM images of 100-nm-diameter Tetraspeck beads (Figure 6—figure supplement 1B-C). The enhanced resolutions of both multicolor 3D-SIM and ODT were laterally sub-200 nm and axially nearly sub-400 nm, respectively (Figure 6—figure supplement 1D).

To investigate the protein compositions of along CAR IS, we used CAR-G4S-mCherry-expressing CART19 cells stained with lysotracker dyes and hCD19-G4S-Zsgreen-expressing K562-CD19 cells. To verify the successful staining of lytic granules before 3D-SIM, we employed wide-field deconvolution fluorescence microscopy and confirmed the rapid dynamics of lytic granules coalescing into the IS (Figure 7—figure supplement 1 and see Figure 6—video 1). We then chemically fixed stable CART19/K562-CD19 conjugates and imaged them using 3D-SIM and ODT for high-resolution multi-protein composition analysis.

We first checked whether additional training is needed. 3D-SIM images provided clearer 3D features of the IS between CART19 and K562-CD19 cells than wide-field microscopy (Figure 7A). Importantly, our DeepIS framework demarcated the interface regions in close proximity to the overlapping areas of CAR and CD19 proteins. From this result, we conclude that the current framework is sufficient for defining the 3D IS areas without additional training with high-resolution data. Next, in order to validate the protein compositions in the CAR IS, we quantitatively analyzed the 3D-SIM images of CAR, CD19, and lytic granules along the IS demarcated by DeepIS (Figure 7B). Consistent with the previous report (https://doi.org/10.1073/pnas.1716266115), the protein compositions of CAR exhibited asymmetric and granular distributions along the CAR IS. From the line and surface profiles of the protein signals, we additionally observed the highest correlations of CAR with lytic granules, as well as colocalizations of CD19 proteins (Figure 7C). Interestingly, the total surficial protein densities obtained by ODT exhibited both correlated and uncorrelated clustered regions with the dense multi-protein clusters. Since ODT quantifies the total protein concentration, the uncorrelated signals are highly likely to imply the presence of clusters of other dominant proteins such as F-actin, Lck, and supramolecular attack particles. Taken together, these results suggest that our DeepIS method based on ODT can be used to define IS area with high accuracy, and also provides collective information about the distribution of total proteins within the IS which may not be easily measured by using conventional fluorescence microscopy.

In summary, we believe that the new experimental results address the remaining issues commented by the reviewers and suggest that our present method provides a complementary analytical modality to fluorescence microscopy for assessing CAR IS. We have added these to the main section of the second revised manuscript.

[Editors' note: further revisions were suggested prior to acceptance, as described below.]

Essential revisions:1) In 2014 Dan Davis' lab developed a negative stain method using fluorescent dyes in the media to define the location of the immunological synapse using confocal microscopy. See: Cartwright AN, Griggs J, Davis DM. The immune synapse clears and excludes molecules above a size threshold. Nat Commun. 2014;5:5479. Epub 2014/11/20. doi: 10.1038/ncomms6479. PubMed PMID: 25407222; PMCID: PMC4248232. It should be possible to use this method with 3D SIM or perhaps Airyscan confocal microscopy to define the location of "synaptic cleft" between the Car-T and target with 125 nm resolution. Correlative ODT scored with Deep IS to define the synapse could then be quantitatively compared to the 3D SIM/confocal analysis of the synaptic cleft to objectively assess the accuracy of deep IS to a ground truth measurement of greater resolution. See below discussion for resolution that would make the 3D SIM a reasonable ground truth for the ODT data sets. The reviewers would accept this approach as an alternative to electron microscopy.

Thank you for the comment. We also appreciate the suggestion, to which we fully agree. As suggested, we conducted an additional experiment based on a negative stain method with 3D-SIM (Figure 6—figure supplement 2). The results show that the 3D SIM data can be used as a reasonable ground truth for the ODT data sets.

We used the FITC-labelled dextran with two varying hydrodynamic diameters of 4 and 54 nm. We added dextran solutions to the chemically fixed conjugate of K562-CD19 and T cells expressing CAR-G4S-mCherry for a negative stain method.

The 3D-SIM images showed that the synaptic cleft in the vicinity of the CAR protein was only visible with the smaller dextran (Figure 6—figure supplement 2A). The fluorescence intensity of dextran relative to the background solution quantified the size-dependent access of dextran into the CAR IS (Figure 6—figure supplement 2B). A two-tail unpaired Wilcoxon test indicated that the fluorescence peak signals were significantly stronger for the smaller dextran (Figure 6—figure supplement 2C). The overall results suggest that the CAR IS excludes dextran molecules above a size threshold, which is consistent with the previous study by Davis’ lab.

To validate, we then compared the imaged synaptic clefts, the CAR fluorescence, and the IS drawn by DeepIS (Figure 6A-B). The 3D overlaps of the three images confirmed the segmentation accuracy of our method (Figure 6C). We plotted the signals across the IS to quantitatively assess the segmentation accuracy (Figure 6—figure supplement 2D). The deviations of the IS from the peak intensities of the CAR and the synaptic cleft were within 200 and 600 nm, respectively. The analysis suggested that the IS retrieved by DeepIS reflected the IS boundary closer to CAR T cells, whose accuracy was comparable to the resolution limit of ODT. Moreover, consistent with the previous report using confocal fluorescence microscopy, the lateral full-width half-maximum of the synaptic cleft was measured to be 894 nm, which may imply the presence of a spatial cavity larger than tens of nanometers across the IS.

Altogether, the additional experiment with a negative stain method validated our proposed method and provided more insights about the CAR IS.

2) For the resolution definition in ODT, you appear to be using the Nyquist sampling period as the imaging resolution. To resolve structures, we need the sample to be separated by at least 2 Nyquist sampling periods. Therefore, the optical resolution should be 2x of the Nyquist resolution you have used in the manuscript. This definition is also consistent with the Abbe resolution definition. Therefore, the lateral resolution in ODT should be λ/2NA which is more than 200 nm not 125 nm. Similarly, the axial resolution should be corrected. If you insist on using Nyquist sampling period as resolution (which seems incorrect), you should clarify it and compare it with the conventionally used Abbe resolution limit. You should also see if you can obtain a 3D standard sample, like an Argolight slide, to directly determine the resolution of the ODT system in a more direct manner.

The reviewers were correct in the definition of resolution in incoherent microscopy. However, unlike incoherent microscopy such as fluorescence microscopy, the definition of resolution in a coherent imaging system such as ODT is not straightforward and varies widely in the community (please also see: doi.org/10.1038/nphoton.2015.279).

The empirical spatial resolution of ODT has been known to be between the Nyquist sampling period and the Abbe criterion (doi.org/10.1364/OPTICA.4.000460), so the Nyquist sampling period is used to set the lowest bound of the resolution. This holds with our previous comparison of the theoretical and experimental resolutions in Figure 6 (also supplement Figure 1(e)) using a 100-nm-diameter Tetraspeck bead as a 3D standard sample. In the manuscript, we defined the experimental resolution as the full width at half maximum, and the experimentally determined values were between the Nyquist sampling period and the Abbe resolution, confirming our statement. To avoid confusion, we have included additional explanations regarding the spatial resolution of ODT.

3) Aside from resolution, there is also a concerned about the refractive index (RI) sensitivity of ODT. When structures have very small (RI) contrast, it becomes a sensitivity issue more than a resolution issue, e.g., ODT is not able to detect many organelles or granules in the cells even when their diameters are larger than 200 nm (the resolution limit). Could the authors provide an estimation of RI contrast between the membranes and estimate whether they have the detection sensitivity for detecting granule release events.

Thank you for raising an important point. To determine the RI sensitivity, we have additionally analyzed a RI tomogram in the lytic granule transport data (Figure 7—figure supplement 2). We compared the RI statistics of a background, cell volume, IS defined by DeepIS, and lytic granules (defined by the overlapping region between a RI cell volume and a fluorescence image with higher than 20% intensity) (Figure 7—figure supplement 2A). The histograms of the four corresponding regions indicated significant differences (Figure 7—figure supplement 2B).

The experimental signal-to-background ratios of the RI contrast were larger than 30. Among the signal groups, the IS and the lytic granules exhibited the significantly large RI differences. The low RI value of the IS was due to the presence of the synaptic cleft, and the higher RI value of the lytic granules was due to the protein clustering. The large RI variance of the lytic granules implies the low chemical specificity of the RI contrast, which could be overcome with correlative fluorescence imaging as previously discussed in the manuscript. The results suggested that a combination of ODT and fluorescence imaging could provide sufficient sensitivity and specificity for detecting granule release events at the IS.

With regard to RI, you continue to interpret the RI in terms of proteins density and this this seems overly simplistic as area within the resolution limit of the interface will contain many types of structures. So this interpretation should be toned down to more of a hypothetical argument about average macromolecule density given convolved RI of lipids, proteins and carbohydrates in the "synapse".

We agree with the viewpoint. We have accordingly toned down several arguments in the revised manuscript.

[Editors' note: further revisions were suggested prior to acceptance, as described below.]

There are 3 remaining issues that need to be addressed before acceptance, as outlined below:1) The Results still state that “lymphocytes contain lipid-rich environment localized mostly on a 4-nm-thick membrane site, whose size is beyond optical resolution and implies a negligibly small amount of lipid molecules compared with proteins.” The reviewers and editor agree that the membrane lipid and protein contributions to the refractive index in relation to ODT is likely to be small for the reason you state, with the major contribution to the pixel intensity being cytoplasmic protein and cortical cytoskeleton. Can you rewrite this passage to clarify the situation that the issue is not so much lipid vs protein, but sub resolved membrane vs the relatively voluminous adjacent cytoplasm.

Agree. We have clarified the sentence, as suggested.

2) The error of synaptic cleft determination is on the order of 0.5 µm and this is a serious limitation currently as it suggest the ability of an expert trainer to correctly identify the synaptic cleft from ODT images is poor. Can you discuss the idea training using the negative staining with the smallest dextran as a ground truth might further improve the power of the approach by eliminating the training error. Then if the ODT contains subtle information that the algorithm can can be trained to exploit, this information will be delivered with not bias other than the resolution limit. Then you would need much more correlative negative stain-ODT for training set and a distinct testing set to determine if there is an improvement. We are not asking you to do this, just to discuss it as a possible way to improve the method in the future.

Thank you for the suggestion. In the Discussion of the revised manuscript, we have elaborated possible ways to enhance the precision of synaptic cleft determination, including the idea training using the negative staining with the smallest dextran or using correlative electron micrographs as a ground truth.

3) The statement in the Abstract that you can't do long term 3D fluorescence microscopy has been made false in recent years by lattice light sheet microscopes. While these are not widely available, they will be in future years based on the new Zeiss LLSM7 release. Please rewrite the Abstract to acknowledge lattice light sheet microscopy as a competing technology and perhaps discuss how ODT with analysis trained to identify the synapse will offer complementary information to LLSM.

Agree. As suggested, we have mentioned lattice light sheet microscopes in the Introduction of the revised manuscript, due to the Abstract length limitation.